# Subcellular second messenger networks drive distinct repellent-induced axon behaviors

Sarah Baudet [1], Yvrick Zagar[1], Fiona Roche[1], Claudia Gomez-Bravo[1], Sandrine Couvet[1], Johann Bécret[1], Morgane Belle[1], Juliette Vougny[1], Sinthuya Uthayasuthan[1], Oriol Ros [1,2,3] & Xavier Nicol [1,3] ✉

Second messengers, including cAMP, cGMP and $Ca^{2+}$ are often placed in an integrating position to combine the extracellular cues that orient growing axons in the developing brain. This view suggests that axon repellents share the same set of cellular messenger signals and that axon attractants evoke opposite cAMP, cGMP and $Ca^{2+}$ changes. Investigating the confinement of these second messengers in cellular nanodomains, we instead demonstrate that two repellent cues, ephrin-A5 and Slit1, induce spatially segregated signals. These guidance molecules activate subcellular-specific second messenger crosstalk, each signaling network controlling distinct axonal morphology changes in vitro and pathfinding decisions in vivo.

Second messengers, including cyclic nucleotides (cAMP and cGMP) and calcium ($Ca^{2+}$), are key signaling molecules involved in a wide range of cellular pathways. Although diffusing freely in aqueous buffers, the mechanisms enabling them to achieve specificity for their many downstream cellular processes rely on the compartmentation of these signaling molecules[1,2]. The compartmentation of $Ca^{2+}$ has been identified in a range of cell types with a variety of subcellular locations. In developing neurons, $Ca^{2+}$ transients have been imaged in growth cones, at the tip of extending axons. Slow transients covering the entire growth cone have been imaged, whereas $Ca^{2+}$ elevations of smaller spatial spread have been identified in filopodia and at cell adhesion sites[3–5]. cAMP nanodomains have been described in a variety of forms including biomolecular condensates of high concentration of this second messenger, local compartments with low cAMP amounts or nanodomains containing receptor-specific signaling units[6–8]. In developing neurons, lipid raft-restricted cAMP signals regulate axon pathfinding[9]. The subcellular compartmentation of cGMP has been less investigated but recent studies have identified distinct sub-membrane domains of this second messenger in cardiomyocytes[10]. However, the functional relevance of subcellular second messenger compartments is still elusive.

Cyclic nucleotides and $Ca^{2+}$ pathways are highly interdependent[11,12]. A subset of cAMP and cGMP synthesizing enzymes are $Ca^{2+}$-regulated[13,14].

The degradation enzymes shared by cyclic nucleotides induce a crosstalk between these signaling molecules[15]. For instance, phosphodiesterase 2 is stimulated by cGMP, thus leading to a cGMP-induced hydrolysis of cAMP[16]. Both the extracellular $Ca^{2+}$ influx and the release of intracellular stores are influenced by the concentration of cAMP and cGMP[17]. Thus, interacting signaling has been identified in many cell types using cell-wide approaches[18–21]. How second messenger compartmentation influences the subcellular interactions between their signaling pathways has been scarcely approached.

In the developing nervous system cAMP, cGMP and $Ca^{2+}$ are key molecules for the establishment of a precisely connected neuronal network. Among other cellular processes occurring in developing neurons, cyclic nucleotides and $Ca^{2+}$ are critical regulators of axonal behavior when the growth cone at the axon distal end faces guidance molecules[22–24]. These cues are expressed in the developing nervous systems and enable axons to follow a genetically-defined path that guide them toward their targets where they connect appropriate post-synaptic neurons. They influence the orientation of axon outgrowth by repelling them or promoting axon extension[25]. The influence of second messengers in this process has been mostly investigated using pharmacological approaches that do not enable subcellular manipulations. These investigations highlighted that, like in other cell types, cyclic nucleotides and $Ca^{2+}$ interact to regulate axon pathfinding[26,27]. A few

[1]Sorbonne Université, INSERM, CNRS, Institut de la Vision, F-75012 Paris, France. [2]Present address: Department of Cell Biology, Physiology and Immunology, Universitat de Barcelona, 08028 Barcelona, Catalonia, Spain. [3]These authors contributed equally: Oriol Ros, Xavier Nicol. ✉e-mail: xavier.nicol@inserm.fr

morphologically- or biochemically-defined compartments of the axonal growth cone have been identified as key locations for axon guidance. In developing neurons, filopodia-restricted $Ca^{2+}$ and cAMP signals orient axon outgrowth[5,26] and cAMP signals restricted to the vicinity of lipid rafts are required for ephrinA5-induced retinal axon repulsion[9]. This fraction of the plasma membrane is also required for the impact of other guidance molecules on growing axons, including Semaphorin-3A and Netrin1[28,29]. Overall, these observations led to a model in which second messengers are positioned as integrators of all the molecular cues detected by growing axons, and in which a reversal of the cAMP:cGMP ratio is sufficient to convert axon attraction into repulsion[23,27].

This model suggests that repellent molecules share a common set of second messenger signals, whereas attractants reverse the ratio of cyclic nucleotide concentration. However, not all axon repellents induce the same morphological changes on developing axons, challenging the idea of a common integrative signal based on second messenger overall concentration. For instance, although Slit2 and Semaphorin-3A both repel the axons of dorsal root ganglia neurons, Slit2 induces a rapid elongation of the filopodia before repulsion, whereas axonal growth cones exposed to Semaphorin-3A do not exhibit this striking behavior[30]. This differential behavior suggests that distinct signaling pathways are involved downstream of different axon repellents. We hypothesize that specific second messenger signals (e.g., restricted to different subcellular domains), control this diversity of axonal responses. In vivo, axon repellents from the Slit and ephrin-A families are involved in non-overlapping developmental stages during the pathfinding of retinal ganglion cell (RGC) axons. Whereas Slits are critical for maintaining axons in the optic nerve and tract when retinal axons reach the optic chiasm[31,32], ephrin-As contribute to terminate retinal axon growth and control the position of their terminal arbors within their main targets in the brain, the superior colliculus (SC) and visual thalamus[33,34].

We hypothesize that Slits and ephrin-As initiate distinct and subcellularly-confined second messenger signals that are associated with their specific influence on axon behavior. Here, using a genetically-encoded toolset enabling the subcellular monitoring and manipulation of second messengers, we tested this hypothesis in developing RGC axons. Focusing on repellent axon guidance molecules from two distinct families (ephrin-A and Slit), we provide a comprehensive description of second messenger signals and their subcellular interactions in axons facing these cues. We demonstrate that second messenger signaling is confined in a single membrane compartment for each guidance molecule, but that the cellular domain involved differs from one cue to another. These differences correlate with distinct axonal behaviors in response to each cue. Consistently, manipulating second messengers in the subcellular compartment corresponding to either Slit1 or ephrin-A5 induces axon pathfinding defects matching the role of each of these guidance molecules in vivo. These observations challenge the theory that the signaling pathways of all guidance molecules are globally integrated by a single set of second messenger modulations.

## Results

### Lipid rafts are the seat of ephrin-A5-induced cGMP and $Ca^{2+}$ signals

Since ephrin-A5 leads to a reduction in cAMP concentration restricted to lipid rafts[9], we focused on the same cellular compartment to identify a potential domain where changes in the level of cGMP are confined downstream of this axon guidance molecule. We used $^{T}hPDE5^{VV}$, a cGMP-sensitive FRET biosensor, to monitor the concentration of this second messenger in retinal axons in vitro. A plasmid encoding $^{T}hPDE5^{VV}$ was electroporated in E14.5 mouse retina. Retinal explants from the electroporated retina were cultured and growing axons were imaged while exposed to ephrin-A5. Since $^{T}hPDE5^{VV}$ is distributed

throughout the entire cytosol, it does not enable to identify the source of cGMP. To this aim, the biosensor was co-electroporated with SponGee, a genetically-encoded cGMP scavenger[35]. Variants of SponGee restricted to lipid rafts (Lyn-SponGee), or to the non-raft fraction of the plasma membrane (SponGee-Kras) are available. The subcellular targeting of Lyn-SponGee relies on a N-terminal fusion of a tandem of palmitoylation-myristoylation sites from Lyn Kinase, whereas SponGee-Kras is restricted to the plasma membrane but excluded from lipid rafts by the C-terminal fusion of a CaaX-polylysine motif derived from K-Ras[35]. The expression of the non-targeted variant of SponGee prevents the elevation of the cGMP concentration independently of its subcellular location, whereas Lyn-SponGee and SponGee-Kras enable to prevent the cGMP changes specifically in lipid rafts or in the non-raft plasma membrane, respectively[35]. When co-expressed with $^{T}hPDE5^{VV}$, the variants of SponGee thus enable to localize the subcellular origin of the cGMP signal (Fig. 1a). When expressed alone, $^{T}hPDE5^{VV}$ detects a transient elevation of cGMP shortly after retinal axons are exposed to ephrin-A5. By contrast, this cGMP elevation is not detected after a sham stimulation (Fig. 1b). This ephrin-A5-induced increase in cGMP concentration is prevented by the cytosolic SponGee and by its lipid raft-targeted variant Lyn-SponGee, but not by the lipid raft-excluded SponGee-Kras, demonstrating that ephrin-A5 exposure leads to an elevation of cGMP in the vicinity of lipid rafts (Fig. 1b).

To evaluate whether the same subcellular compartment is also the seat of $Ca^{2+}$ signals, the $Ca^{2+}$ biosensor Twitch2b was used[36]. A lipid raft-restricted variant of Twitch2b (Lyn-Twitch2b) was engineered. The lipid raft targeting of Lyn-Twitch2b was confirmed by membrane fractionation on a sucrose gradient and was found in the same fractions as the lipid raft marker Caveolin (Supplementary Fig. 1). Similarly, Twitch2b was fused to the Kras targeting sequence (Twitch2b-Kras) that shifts the localization of Twitch2b toward the fractions labeled by the non-raft marker Adaptin, demonstrating that this sensor is not targeted to lipid rafts (Supplementary Fig. 1). These two biosensors enable the direct visualization of subcellular $Ca^{2+}$ signals (Fig. 1c), following a strategy recently used with another $Ca^{2+}$ biosensor[37]. Retinal axons expressing either Twitch2b (not targeted) or Lyn-Twitch2b (lipid raft-targeted) exhibited an increase in the frequency of $Ca^{2+}$ transients upon ephrin-A5 exposure (Fig. 1d). These $Ca^{2+}$ transients are characterized by a brief elevation of the $Ca^{2+}$ concentration lasting in the range of 10 s and resembles the $Ca^{2+}$ transients previously described in the growth cones of developing axons in vitro and in vivo[4,26,38]. By contrast, the Kras-targeted variant of Twitch2b did not detect any ephrin-A5-induced change in $Ca^{2+}$ signals (Fig. 1d). Thus, ephrin-A5 induces an increase in the frequency of $Ca^{2+}$ transients that are detected in lipid rafts but not outside of this membrane compartment.

Overall, this set of experiments identifies lipid rafts as a subcellular compartment concentrating the second messenger signals induced by ephrin-A5 in developing axons. It includes a cAMP reduction[9], a cGMP elevation and an increase in the frequency of $Ca^{2+}$ transients.

### Slit-induced cAMP, cGMP and $Ca^{2+}$ signals are excluded from lipid rafts

Since the current model of second messenger signaling involved in axon pathfinding places these signaling molecules at the crossroads of many guidance cues, we evaluated whether another retinal axon repellent (Slit1) induces second messenger signals in the same subcellular compartment as ephrin-A5. Using similar approaches as the ones described above, we characterized the subcellular features of cAMP, cGMP and $Ca^{2+}$ signals induced by Slit1.

cAMP was monitored using the FRET biosensor H147[39], for which a lipid raft-targeted (Lyn-H147) and a lipid-raft excluded (H147-Kras) variant are available[9] (Fig. 2a). Using the cAMP sensor H147 without subcellular targeting, we found that Slit1 induces an overall reduction in cAMP concentration in developing retinal axons (Fig. 2b). This

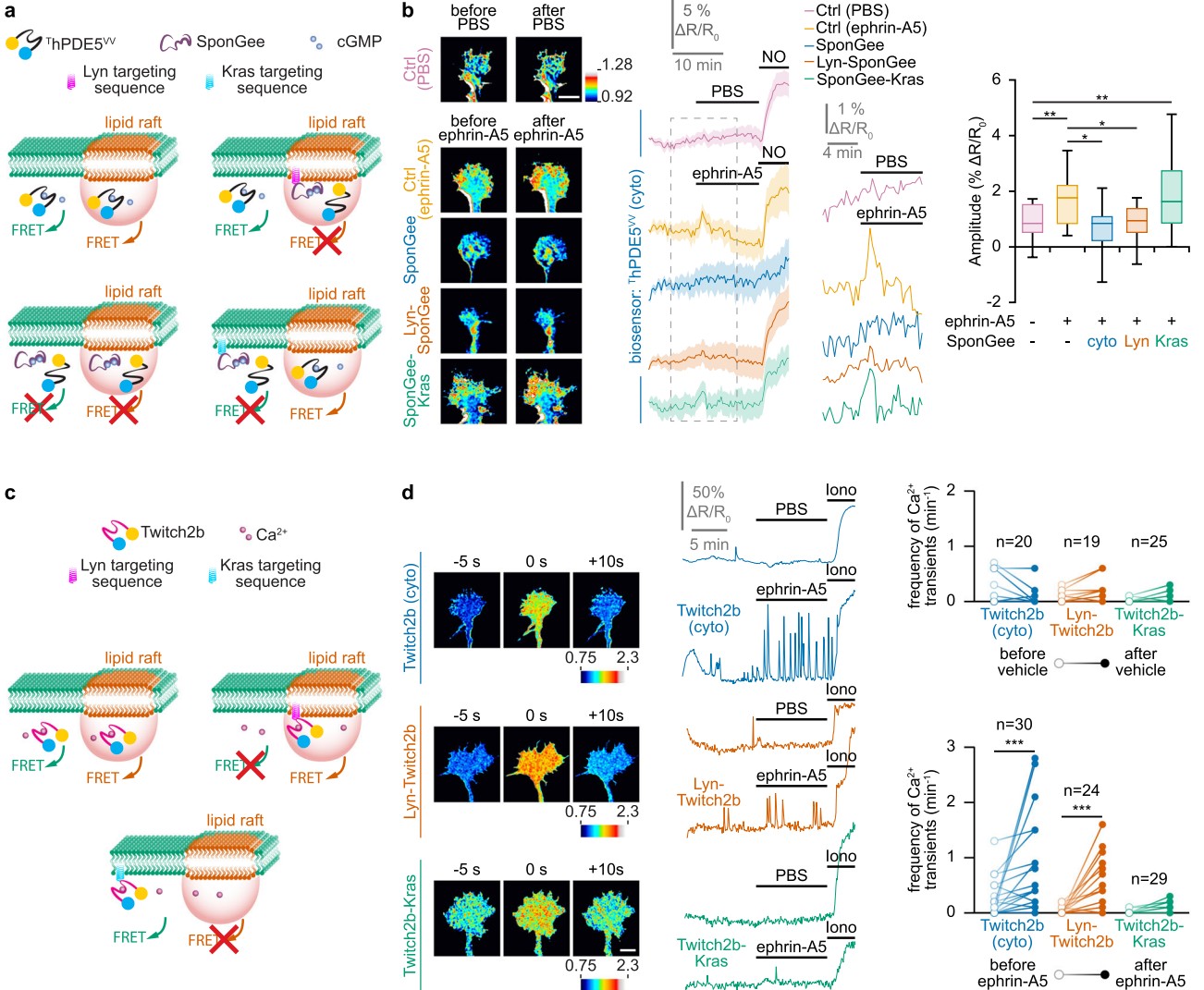

**Fig. 1 | Ephrin-A5 induces an elevation of cGMP and an increase in the frequency of Ca²⁺ transients in lipid rafts. a** Strategy to identify the cGMP source. The cGMP biosensor ᵀhPDE5ᵛᵛ is expressed in RGC axons alone (top left) or together with either the non-targeted cGMP scavenger SponGee (bottom left) or its lipid raft-targeted (Lyn-SponGee, top right) or -excluded (SponGee-Kras, bottom right) variants. When expressed alone, ᵀhPDE5ᵛᵛ monitors cGMP from the entire cytoplasm. SponGee prevents the sensor to report changes in cGMP concentration. Lyn-SponGee or SponGee-Kras lead ᵀhPDE5ᵛᵛ to monitor cGMP changes from all cellular compartments excluding the vicinity of either lipid rafts or the non-raft plasma membrane, respectively. **b** After ephrin-A5 exposure, ᵀhPDE5ᵛᵛ alone or co-expressed with SponGee-Kras monitors a cGMP elevation. By contrast, when co-expressed with SponGee or Lyn-SponGee, the FRET signal is not affected by ephrin-A5, similarly to vehicle (PBS)-exposed axons. A nitric oxide (NO)-induced cGMP elevation at the end of each recording ensures biosensor functionality and axon viability. Right traces: magnification of the dashed line-enclosed portion of the left traces. Image color-code: from low cGMP (blue) to high cGMP (red/white). Traces:

mean ± s.e.m. Box-and-whisker plot elements: median, upper and lower quartiles, 10th and 90th percentiles. *$P < 0.05$; **$P < 0.01$; Kruskal–Wallis test followed by Dunn's post-hoc test. Scale bar, 10 μm. **c** Strategy to identify local changes in Ca²⁺ concentration. The Ca²⁺ sensor Twitch2b (top left), its lipid raft-targeted (top right) or -excluded (bottom) variants are expressed in RGCs. They report Ca²⁺ changes from all cytosolic sources, lipid rafts and the non-raft fraction of the plasma membrane, respectively. **d** An elevation in the Ca²⁺ transient frequency is detected by Twitch2b after ephrin-A5 but not after vehicle (PBS) exposure. This observation is reproduced with the lipid raft-targeted Twitch2b, but not when using its lipid raft-excluded equivalent. An ionomycin (iono)-induced Ca²⁺ elevation at the end of each recording ensures biosensor functionality and axon viability. Images illustrate a detected Ca²⁺ transient. Color-code: from low Ca²⁺ (blue) to high Ca²⁺ (red/white). Representative traces and individual data points (n indicated on the graphs) are shown. *** $P < 0.001$; two-tailed Wilcoxon test. Scale bar, 10 μm. Source data, number of replicates and P values are provided as a Source Data file.

reduction was also detected by a lipid raft-excluded variant of this sensor (H147-Kras), whereas axons expressing the lipid raft-targeted Lyn-H147 exhibited no change in the FRET signal, reflecting a stable cAMP concentration (Fig. 2b). This highlights that Slit1 modulates cAMP in a compartment that is close to plasma membrane but excluded from lipid rafts, contrasting with the ephrin-A5-induced cAMP modulation[9].

cGMP concentration was monitored in Slit1-stimulated retinal axons using ᵀhPDE5ᵛᵛ. An elevation in cGMP was detected upon Slit1 exposure, mimicking the ephrin-A5-dependent signals (Fig. 3a).

However, Slit1-induced cGMP increase was prevented by the lipid raft-excluded scavenger SponGee-Kras, whereas the lipid raft-targeted equivalent Lyn-SponGee was unable to reduce this cGMP elevation (Fig. 3a). This experiment demonstrates that Slit1 controls the cGMP concentration in the vicinity of the non-raft plasma membrane, in contrast to ephrin-A5.

Ca²⁺ concentration was imaged in Slit1-exposed retinal growth cones. Similar to ephrin-A5, an elevation of the frequency of Ca²⁺ transient was detected when using the non-targeted biosensor Twitch2b (Fig. 3b). In contrast to ephrin-A5, the Twitch2b variant

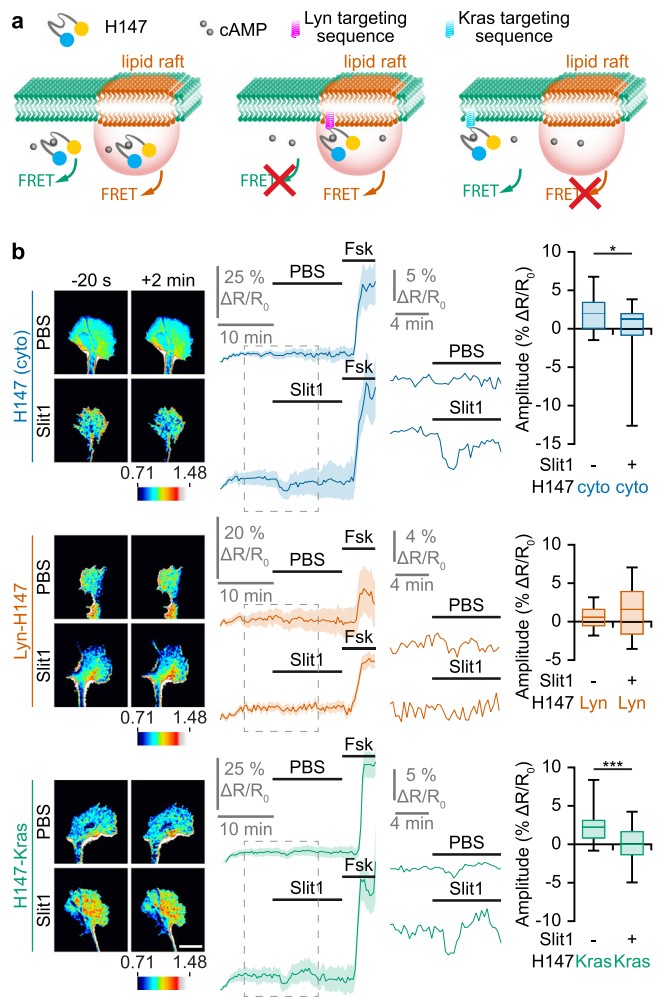

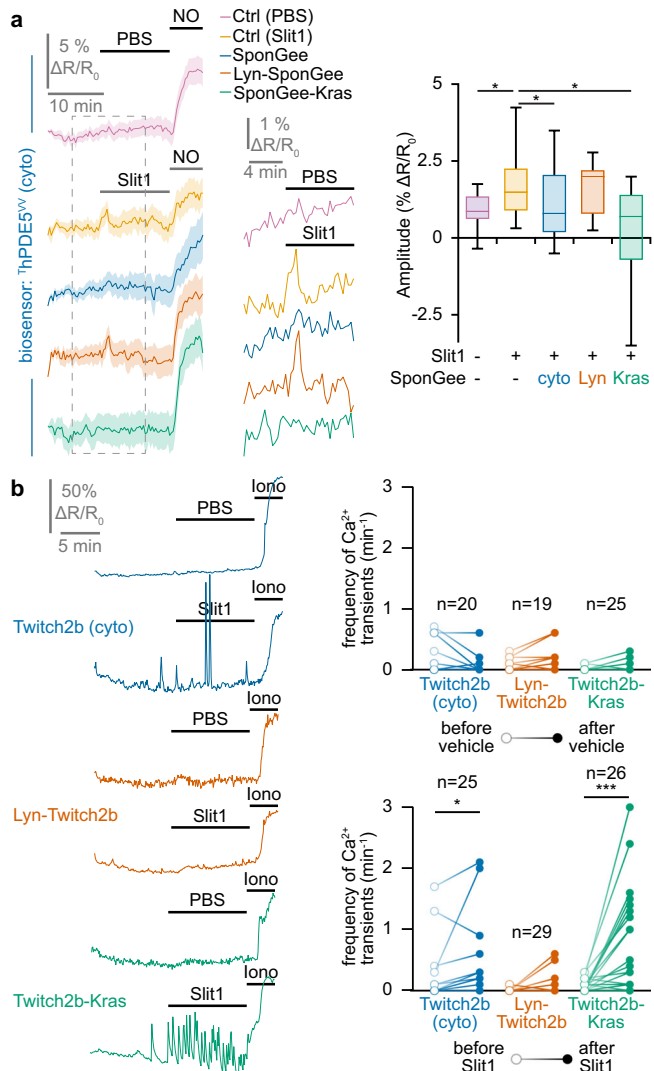

**Fig. 2 | Slit1 induces a cAMP reduction in the vicinity of the plasma membrane, outside lipid rafts. a** Strategy to identify local changes in cAMP concentration. The cAMP FRET sensor H147 (left), its lipid raft-targeted (middle) or -excluded (right) variants are expressed in RGCs. They report cAMP changes from anywhere in the cytosol, from lipid rafts and from the non-raft fraction of the plasma membrane, respectively. **b** A reduction in the cAMP concentration is detected by the biosensor H147 after Slit1 exposure but not after vehicle (PBS) addition to the culture medium. This observation is reproduced with the lipid raft-excluded H147 (H147-Kras), but not when using its lipid raft-targeted equivalent (Lyn-H147). A forskolin (Fsk) stimulation leading to a cAMP elevation was achieved at the end of each recording to verify the functionality of the biosensor and the viability of the axon. The portion of the left traces enclosed in the dashed rectangles is shown magnified in the right part of the panel. Images illustrating the change in the FRET ratio between before (−20 s) and after (+2 min) the PBS or Slit1 stimulation are color-coded from low blue to high red/white. Traces: mean ± s.e.m. Box-and-whisker plot elements: median, upper and lower quartiles, 10th and 90th percentiles. *$P < 0.05$; ***$P < 0.001$; two-tailed Mann–Whitney test. Scale bar, 10 µm. Source data, number of replicates and $P$ values are provided as a Source Data file.

**Fig. 3 | Slit1 induces a cGMP elevation and an increase in the frequency of $Ca^{2+}$ transients in the vicinity of the plasma membrane, outside lipid rafts. a** When retinal axons are exposed to Slit1, $^{T}hPDE5^{VV}$ alone or co-expressed with Lyn-SponGee monitors an elevation of cGMP. By contrast, when co-expressed with SponGee or SponGee-Kras, the FRET ratio is not affected by Slit1, similarly to axons that are not exposed to Slit1 and express $^{T}hPDE5^{VV}$. A nitric oxide (NO) stimulation leading to a cGMP elevation was achieved at the end of each recording to verify the functionality of the biosensor and the viability of the axon. The portion of the left traces enclosed in the dashed rectangles is shown magnified in the right part of the panel. Traces: mean ± s.e.m. Box-and-whisker plot elements: median, upper and lower quartiles, 10th and 90th percentiles. **b** An elevation in the frequency of $Ca^{2+}$ transients is detected by Twitch2b after Slit1 exposure but not after vehicle (PBS) addition to the culture medium. This elevation is reproduced with the lipid raft-excluded Twitch2b, but not when using its lipid raft-excluded equivalent. An ionomycin (iono) stimulation leading to a $Ca^{2+}$ elevation was achieved at the end of each recording to verify the functionality of the biosensor and the viability of the axon. Representative traces and individual data points are shown. The number of quantified axons is indicated on the graphs. **a,b** *$P < 0.05$; *** $P < 0.001$; **a** Kruskal–Wallis test followed by Dunn's post-hoc test, **b** two-tailed paired Wilcoxon test. Source data, number of replicates and $P$ values are provided as a Source Data file.

excluded from lipid rafts (Twitch2b-Kras), but not its Lyn-targeted equivalent (raft-targeted), also detected this $Ca^{2+}$ signal, demonstrating that, like for cyclic nucleotides, Slit1-induced $Ca^{2+}$ signals are excluded from lipid rafts (Fig. 3b).

Overall, Slit1 and ephrin-A5 generate a set of second messenger signals that are restricted to distinct subcellular compartments, although they both exhibit a repellent activity on developing retinal axons. Whereas ephrin-A5 induces changes in cAMP, cGMP and $Ca^{2+}$ in the vicinity of lipid rafts, Slit1 leads to modifications in the concentration of the same second messengers near the non-raft fraction of the plasma membrane.

## Subcellular interactions between second messenger signals in developing retinal axons

To further characterize the differences in second messenger signaling downstream of Slit1 and ephrin-A5, we investigated the crosstalks between cyclic nucleotides and $Ca^{2+}$ with subcellular resolution. To this

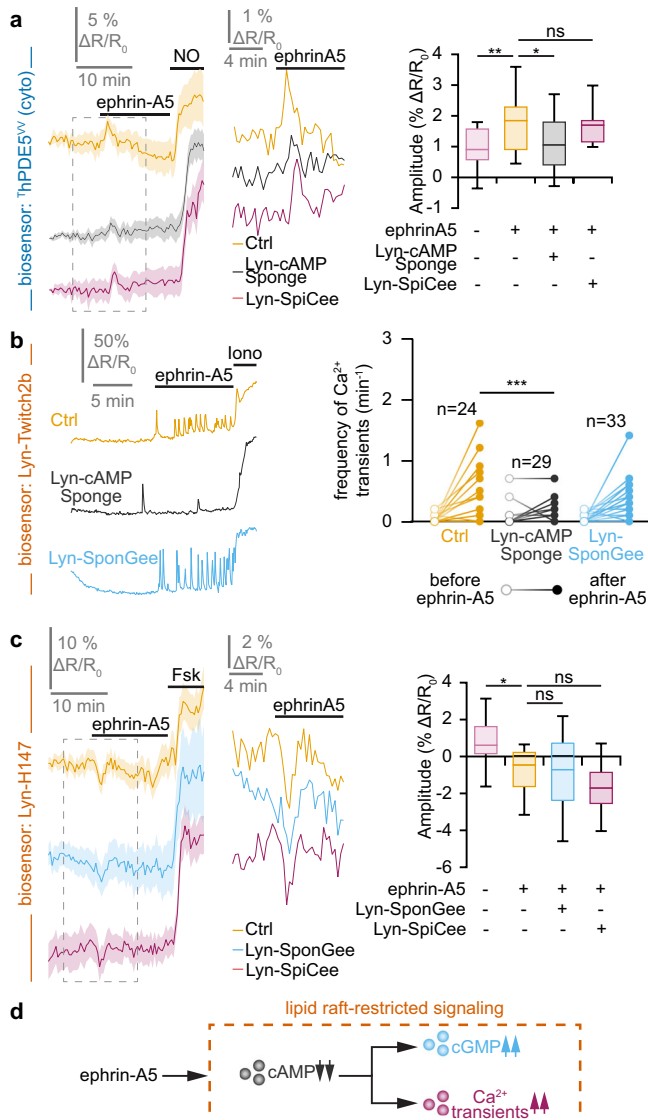

**Fig. 4 | Lipid raft-restricted second messenger network downstream of ephrin-A5. a** The ephrin-A5-induced elevation of the cGMP concentration monitored using $^{T}$hDPE5$^{VV}$ is reduced by preventing local cAMP signals in lipid rafts (Lyn-cAMP Sponge). By contrast, reducing Ca$^{2+}$ downstream signaling in this cellular domain does not affect cGMP changes. A nitric oxide (NO) stimulation was achieved at the end of each recording to verify the functionality of the biosensor and the viability of the axon. **b** The ephrinA5-induced elevation in the Ca$^{2+}$ transient frequency monitored by Lyn-Twitch2b was prevented by scavenging cAMP in lipid rafts (Lyn-cAMP Sponge), whereas altering the downstream signaling of cGMP in this subcellular domain did not impact the ephrin-A5-induced Ca$^{2+}$ transients. An ionomycin (iono) stimulation was achieved at the end of each recording to verify the functionality of the biosensor and the viability of the axon. **c** The lipid raft-restricted cAMP signals induced by ephrin-A5 were monitored using Lyn-H147. The reduction in the cAMP concentration was not affected by preventing the cGMP or Ca$^{2+}$ downstream signaling in the same cellular compartment using Lyn-SponGee and Lyn-SpiCee, respectively. A forskolin (Fsk) stimulation was achieved at the end of each recording to verify the functionality of the biosensor and the viability of the axon. **d** Overall model of the lipid raft-restricted second messenger network downstream of ephrin-A5. Exposing growth cones to this axon guidance molecule leads to a combined modulation of cyclic nucleotides and Ca$^{2+}$ that is restricted to the vicinity of lipid rafts. This network is characterized by a non-reciprocal influence of cAMP on cGMP and Ca$^{2+}$ signaling. **a,c** The portion of the left traces enclosed in the dashed rectangle in is shown magnified in the right part of the panel. Traces: mean ± s.e.m. Box-and-whisker plot elements: median, upper and lower quartiles, 10th and 90th percentiles. **b** Representative traces and individual data points are shown. The number of quantified axons is indicated on the graphs. **a–c** * $P < 0.05$; ** $P < 0.01$; *** $P < 0.001$; Kruskal–Wallis test followed by Dunn's post-hoc test. Source data, number of replicates and $P$ values are provided as a Source Data file.

Buffering cAMP with Lyn-cAMP sponge was sufficient to prevent both the elevation of cGMP detected by $^{T}$hPDE5$^{VV}$ and the increase in Ca$^{2+}$ transient frequency monitored by Lyn-Twitch2b (Fig. 4a,b). By contrast, scavenging cGMP or Ca$^{2+}$ in lipid rafts with Lyn-SponGee or Lyn-SpiCee did not affect the ephrin-A5-induced signals for either of the other second messengers (Fig. 4a–c). Thus, cAMP is positioned upstream of cGMP and Ca$^{2+}$ in lipid rafts of retinal growth cones exposed to ephrin-A5 (Fig. 4d).

In contrast to ephrin-A5, Slit1 modulates cyclic nucleotides and Ca$^{2+}$ in the non-raft fraction of the plasma membrane. Accordingly, $^{T}$hPDE5$^{VV}$, Twitch2b-Kras and H147-Kras were used to evaluate the crosstalk between Slit1-induced second messenger signals. Buffering cAMP changes in the non-raft plasma membrane with cAMP Sponge-Kras did not affect the increase in cGMP but reduced the elevation of the Ca$^{2+}$ transient frequency induced by Slit1 (Fig. 5a,b). Scavenging cGMP outside lipid rafts with SponGee-Kras prevented both the Ca$^{2+}$ and cAMP changes in retinal axons (Fig. 5b,c). Preventing Ca$^{2+}$ signaling in the non-raft membrane by expressing SpiCee-Kras precluded the elevation of cGMP concentration but did not affect the reduction in cAMP (Fig. 5a,c). Thus, Slit1 induces the activation of a complex signaling network intermingling cyclic nucleotides and Ca$^{2+}$ signaling. The relationships between the signaling molecules involved in this network are illustrated in Fig. 5d.

Overall, we demonstrate that not only are second messenger modulations downstream of ephrin-A5 and Slit1 confined to distinct subcellular compartments, but also that the signaling networks formed by these molecules differ downstream of each of these axon guidance cues.

### Domain-specific signals are required for axon repulsion

To determine whether the lipid raft-restricted and -excluded second messenger signals are required for ephrin-A5- and Slit1-induced retraction, the behavior of retinal axons exposed to these guidance molecules was evaluated. RGC axons were grown in vitro after electroporation of the retina with either mRFP or one of the cytosolic or subcellular compartment-targeted scavengers (SpiCee, SponGee or

aim, we benefited from a set of genetically-encoded scavengers targeting cyclic nucleotides and Ca$^{2+}$. cAMP Sponge enables to buffer cAMP in living cells[40] and is available as lipid raft-targeted (Lyn-cAMP Sponge) and -excluded (cAMP Sponge-Kras) variants[9]. SponGee and its variants enable the subcellular manipulation of cGMP[35]. SpiCee is a Ca$^{2+}$ scavenger that has been fused to the lipid raft-targeted and -excluded sequences (Lyn-SpiCee and SpiCee-Kras, respectively)[41]. The ability of the targeted scavengers to prevent second messenger signals was verified by the co-expression of a given scavenger with the corresponding biosensors: $^{T}$hPDE5$^{VV}$ for cGMP (Fig. 1b; Fig. 3a) or the targeted variants of H147 and Twitch2b for cAMP and Ca$^{2+}$, respectively (Supplementary Fig. 2).

To evaluate the interactions between the second messenger signals, RGCs expressing a subcellular-specific biosensor sensitive to second messenger A were co-electroporated with a scavenger preventing the downstream signaling of second messenger B and exposed to either Slit1 or ephrin-A5. For instance, Lyn-Twitch2b was co-expressed with Lyn-cAMP Sponge in axons exposed to ephrin-A5 to determine whether lipid raft-specific cAMP signaling influences the ephrin-A5-induced elevation in the frequency of Ca$^{2+}$ transients in the same subcellular compartment.

Since all second messenger signals detected downstream of ephrin-A5 were found in lipid rafts, we investigated the crosstalk between these signaling molecules within this cellular compartment.

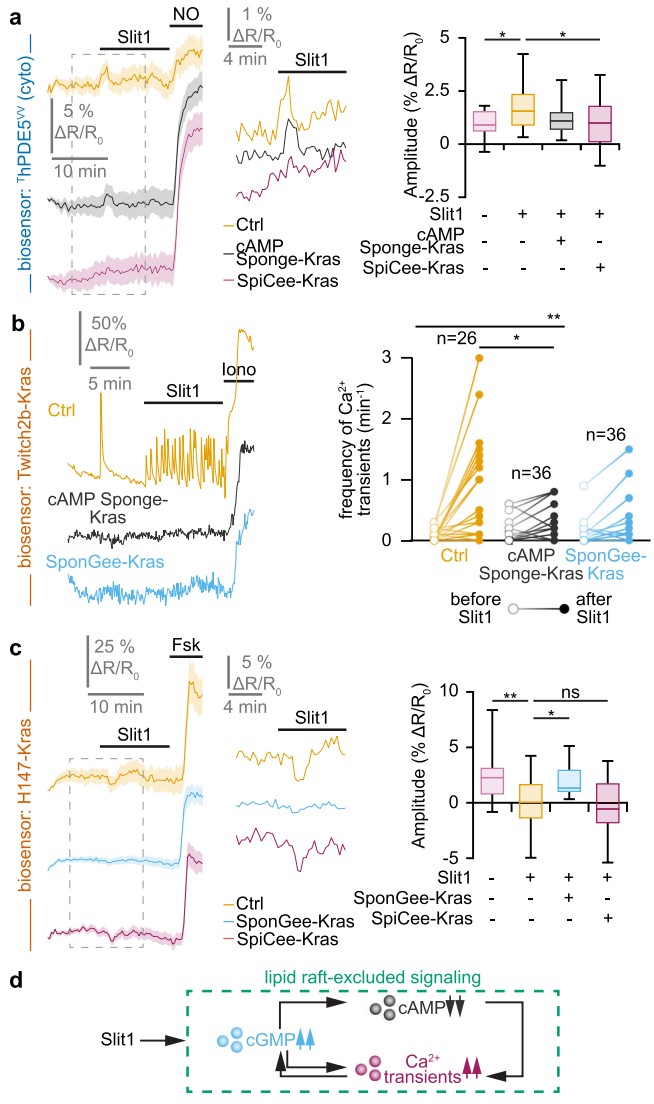

**Fig. 5 | Lipid raft-excluded second messenger network downstream of Slit1. a** The Slit1-induced elevation in cGMP imaged using ᵀhDPE5ᵛᵛ is prevented by the blockade of $Ca^{2+}$ signaling next to the non-raft plasma membrane (SpiCee-Kras). By contrast, blocking cAMP signaling outside lipid rafts (cAMP Sponge-Kras) does not affect the Slit1-induced cGMP changes. A nitric oxide (NO) stimulation was achieved at the end of each recording to verify the biosensor functionality and the axon viability. **b** The elevation in the $Ca^{2+}$ transient frequency induced by Slit1 outside lipid rafts and recorded using Twitch2b-Kras is prevented by scavenging either cAMP or cGMP outside lipid rafts (cAMP Sponge-Kras or SponGee-Kras, respectively). An ionomycin (iono) stimulation was achieved at the end of each recording to verify the biosensor functionality and the axon viability. **c** The juxta-membrane lipid raft-excluded cAMP signals induced by Slit1 were monitored using the biosensor H147-Kras. The reduction in the cAMP concentration is dampened by preventing cGMP downstream signaling in the same cellular compartment using SponGee-Kras, but not by reducing $Ca^{2+}$ signaling with SpiCee-Kras. A forskolin (Fsk) stimulation was achieved at the end of each recording to verify the biosensor functionality and the axon viability. **d** Overall model of the juxta-membrane lipid raft-excluded second messenger network downstream of Slit1. Exposing growth cones to this axon guidance molecule leads to a combined modulation of cyclic nucleotides and $Ca^{2+}$ that is restricted to the vicinity of the non-raft domain of the plasma membrane. This network is characterized by complex interactions including a cAMP influence on $Ca^{2+}$, a control of cGMP elevation by $Ca^{2+}$ transients and cGMP influencing both cAMP and $Ca^{2+}$ signals. **a,c** The portion of the left traces enclosed in the dashed rectangle in is shown magnified in the right part of the panel. Traces: mean ± s.e.m. Box-and-whisker plot elements: median, upper and lower quartiles, 10th and 90th percentiles. **b** Representative traces and individual data points are shown. The number of quantified axons is indicated on the graphs. **a–c** * $P < 0.05$; ** $P < 0.01$; *** $P < 0.001$; Kruskal–Wallis test followed by Dunn's post-hoc test. Source data, number of replicates and $P$ values are provided as a Source Data file.

targeted to the non-raft plasma membrane are expressed (Fig. 6c–e). By contrast, Lyn-SpiCee, Lyn-SponGee and Lyn-cAMP sponge are not able to prevent the Slit1-induced growth cone collapse of retinal axons (Fig. 6c–e), demonstrating that the second messenger signals required for the Slit1-induced collapse of retinal axons are restricted to the non-raft domain of the plasma membrane. This conclusion is confirmed by the unaltered collapse of growth cones expressing variants of SpiCee-Kras, SponGee-Kras and cAMP Sponge-Kras that carry point mutations preventing their ability to bind $Ca^{2+}$, cGMP and cAMP, respectively (mut SpiCee-Kras, mut SponGee-Kras and mut cAMP Sponge-Kras, Supplementary Fig. 3b)

Altogether, we show that the cGMP, cAMP and $Ca^{2+}$ signals inducing the collapse of the growth cone are confined to lipid rafts when the repulsion is driven by ephrin-A5, whereas the compartment involved is the non-raft fraction of the plasma membrane downstream of Slit1.

## Slit1 and ephrin-A5 induce distinct morphological changes of RGC growth cones
The collapse assay provides a coarse characterization of axonal repulsion. Since ephrin-A5- and Slit1-induced growth cone collapse relies on second messenger signaling in distinct cellular compartments, these axon guidance molecules might induce axon repulsion with distinct features. To evaluate whether the response of retinal growth cones exposed to ephrin-A5 diverges from the effect of an exposure to Slit1, the behavior of living axons facing these guidance molecules was monitored during 20 min after adding the axon repellent to the culture medium. Control axons growing in a medium supplemented with PBS continue to extend. By contrast, ephrin-A5-exposed axons quickly collapse and retract, with a fast backward movement (Fig. 7; Supplementary Movie 1). Axons exposed to Slit1 exhibit a different behavior: when facing Slit1, they stop growing and collapse with little or no retraction in the 20 min following the addition of Slit1 to the culture medium (Fig. 7; Supplementary Movie 1). This differential behavior matches the role of ephrin-A5 and Slit1 during the

cAMP Sponge). When exposed to repellents in vitro, axonal growth cones undergo a morphological change that is characterized by the loss of their lamellipodium. The ephrin-A5- or Slit1-induced growth cone collapse was evaluated 20 min after RGC axon exposure to the guidance cue, i.e., shortly after the detected second messenger signals. The second messenger buffers did not affect the morphology of retinal growth cones that were not exposed to Slit1 or ephrin-A5 (Supplementary Fig. 3a).

Ephrin-A5 induces the collapse of mRFP-expressing growth cones. The fraction of collapsed axons was reduced when SpiCee, SponGee or their lipid raft-restricted variants were expressed (Fig. 6a,b). By contrast, SpiCee-Kras and SponGee-Kras did not affect the response of retinal axons to ephrin-A5 (Fig. 6a,b). Similarly, Lyn-cAMP sponge, but not cAMP sponge-Kras, was previously reported to prevent the retraction of retinal axons induced by ephrin-A5[9]. By contrast, scavengers carrying point mutations that abolish their ability to bind second messengers did not preclude the collapse of RGC growth cones (Lyn-mut SpiCee and Lyn-mut SponGee, Supplementary Fig. 3b). Overall, this set of experiments demonstrates that the lipid raft-restricted cyclic nucleotide and $Ca^{2+}$ signals detected after exposure to ephrin-A5 are required for axon repulsion.

When exposed to Slit1 for 20 min, mRFP-expressing axons collapse, whereas a higher number of axons with an intact growth cone were observed when SpiCee, SponGee, cAMP sponge or their variant

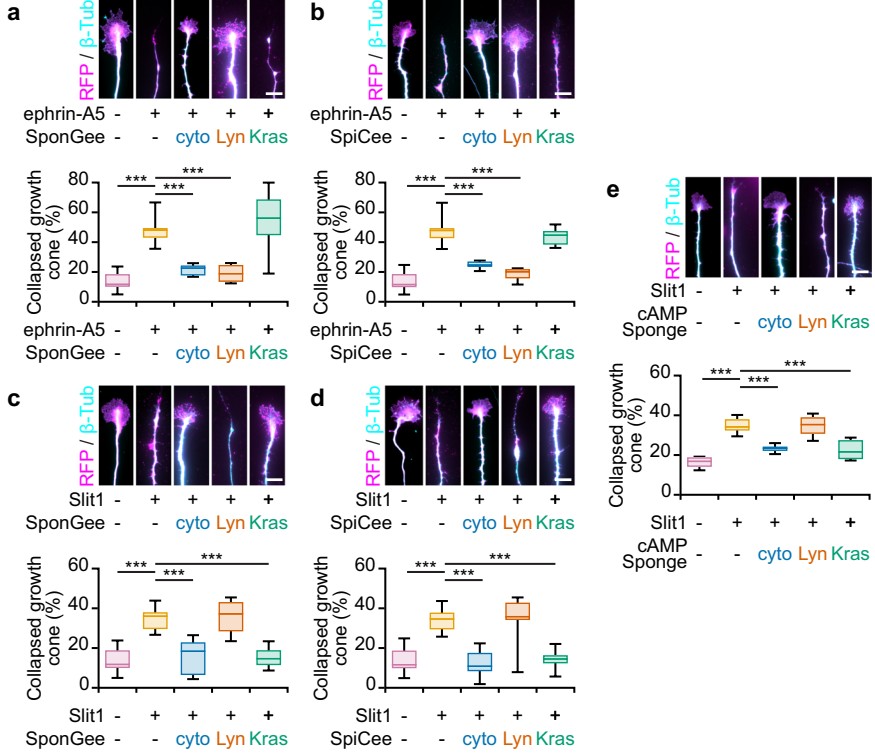

**Fig. 6 | Lipid raft-specific and -excluded scavenging of second messengers prevent the collapse of growth cones induced by ephrin-A5 and Slit-1, respectively. a** Ephrin-A5 induces the collapse of mRFP- and SponGee-Kras-expressing axons, whereas the non-targeted and lipid-raft targeted variants of SponGee (Lyn-SponGee) prevent growth cone collapse. **b** When lacking a targeting sequence or when restricted to lipid rafts, SpiCee prevents the ephrin-A5-induced growth cone collapse, in contrast to the lipid raft-excluded variant of SpiCee (SpiCee-Kras). **c** Slit-1 induces the collapse of mRFP-expressing retinal growth cones. The proportion of collapsing axons is not affected by the expression of Lyn-SponGee but is reduced by SponGee-Kras or the cytosolic SponGee. **d** The collapse of retinal growth cones exposed to Slit-1 is reduced by the expression of SpiCee (not targeted) or its lipid raft-excluded variant (SpiCee-Kras), but not by Lyn-SpiCee. **e** Slit-1-induced growth cone collapse is prevented by cAMP Sponge or cAMP Sponge-Kras, but not by Lyn-cAMP-Sponge. Axons were immunolabeled with a βIII-tubulin and a Ds-Red (mRFP) antibody. The latter reports the expression of SponGee (**a,c**), SpiCee (**b,d**) or cAMP Sponge (**e**). Scale bars, 10 μm. Box-and-whisker plot elements: median, upper and lower quartiles, 10th and 90th percentiles. *** $P < 0.001$; One way ANOVA followed by Dunnett post-hoc test. Source data, number of replicates and $P$ values are provided as a Source Data file.

development of retinal axons. Slit1 is involved in orienting the growth of the axons and in preventing them to exit the corridor formed by the optic nerve, chiasm and tract, but does induce the retraction of retinal axons that have not yet reached their targets. By contrast, ephrin-A5 is expressed in regions of the brain where retinal axons are strongly repelled and exhibit the retraction of branches overshooting the position of their mature terminal arbor[42].

**Cyclic nucleotide modulation imposed in and outside lipid rafts mimic ephrin-A5- and Slit1-induced axon behavior, respectively**

To evaluate whether the difference in the axon behavior induced by ephrin-A5 and Slit1 is controlled by the second messenger signal compartment in distinct cellular domains, we imposed cAMP or cGMP changes in or outside lipid rafts using optogenetics. The light-sensitive adenylyl cyclase bPAC[43] and the light activatable guanylyl cyclase BeCyclOp[44] were used to manipulate the concentration of cAMP and cGMP, respectively. Light pulses were sufficient to induce a transient elevation of cAMP in retinal growth cones expressing the lipid raft-targeted and -excluded Lyn-bPAC and bPAC-Kras[9] (Supplementary Fig. 4). After the end of the light pulses, cAMP concentration dropped below its resting concentration enabling to mimic the detected cAMP signals induced by ephrin-A5 and Slit1 (Supplementary Fig. 4). To ensure minimal excitation of bPAC when imaging the biosensor, these elevations were detected using variants of RFlincA, a red single-wavelength cAMP probe excited using a 561 nm laser line[45]. RFlincA was targeted and excluded from lipid rafts using the Lyn and Kras sequences (Lyn-RFlincA and RFlincA-Kras, respectively). Similarly, we used δFlincG a single-wavelength cGMP indicator to monitor cGMP concentration in BeCyclOp-expressing growth cones[46]. δFlincG was targeted and excluded from lipid rafts using the Lyn and Kras sequences (Lyn-δFlincG and δFlincG-Kras, respectively). Light exposure induced an elevation of cGMP detected by Lyn-δFlincG and δFlincG-Kras in Lyn-BeCyclOp- and BeCyclOp-Kras-expressing growth cones (Supplementary Fig. 4).

Imposing cAMP changes in lipid rafts using Lyn-bPAC led to the collapse of the growth cone followed by the retraction of collapsing growth cones, thus mimicking the axon behavior induced by ephrin-A5. By contrast, changing cAMP concentration outside lipid rafts with bPAC-Kras induced the collapse of the growth cone without subsequent axon retraction, matching the morphological change of Slit1-exposed axons. The growth of control axons that do not express Lyn-bPAC or bPAC-Kras was not affected by light pulses (Fig. 8a). Imposing a cGMP elevation in lipid raft using Lyn-BeCyclOp induced the collapse and subsequent retraction of a limited number of axons while others were insensitive to the stimulation (Fig. 8b). By contrast, most of the axons that experienced a cGMP elevation outside lipid rafts induced by BeCyclOp-Kras stimulation collapsed but did not retract (Fig. 8b). These observations are again in line with the lipid raft-restricted cGMP signals induced by ephrin-A5, that lead to the retraction of the axons, and with the lipid raft-excluded cGMP elevation downstream of Slit1, that causes growth cone collapse without immediate retraction. In addition, the lower number of collapsing growth cones after a lipid raft-restricted cGMP elevation, as compared to a lipid raft-excluded cGMP manipulation or a lipid raft-specific cAMP signal, supports the

second messenger interaction networks drawn in Figs. 4d and 5d, although the requirement of all interactions between second messengers is not directly demonstrated. In lipid rafts, cGMP is placed downstream cAMP and does not influence $Ca^{2+}$ signaling. The

Lyn-BeCyclOp-expressing axons thus experienced only a subset of the ephrin-A5-induced pathways, explaining the low number of collapsing growth cones. By contrast, outside lipid rafts, cGMP influences both cAMP and $Ca^{2+}$ signaling. BeCyclOp-Kras-expressing growth cones are more likely to detect a more complete set of Slit1-dependent signals, thus making them more prone to mimic the Slit1-induced change of behavior.

## Impact of second messenger signaling on the development of retinal axon arbors in vivo

To evaluate whether the second messenger signals confined to or excluded from lipid rafts regulate distinct repulsion behaviors in vivo, the lipid raft-targeted or -excluded version of SponGee or SpiCee was electroporated in utero in the developing retina of E14.5 embryos. Regions of the brain where retinal axon pathfinding relies on Slit1 or ephrin-A5 were analyzed after whole brain clearing and light-sheet imaging. The SC where ephrin-As are critical to shape retinotopic mapping and prevent axons from invading the inferior colliculus, and the optic chiasm where retinal axons are confined within their correct path by Slit1 and Slit2. mRFP-electroporated axons arborize in the SC and form dense terminal arbors at P15. No axonal branches were found in the inferior colliculus, where ephrin-As are highly expressed. By contrast, retinal axons expressing either Lyn-SponGee or Lyn-SpiCee exhibited exuberant branches in the inferior colliculus. In a few cases with sparse electroporation, multiple termination zones for a single axon were found in the SC. These abnormal retinal axon projections were not detected in animals electroporated with the raft-excluded SponGee-Kras and SpiCee-Kras (Fig. 9a). Similar observations were previously reported for Lyn-cAMP Sponge and cAMP Sponge-Kras[9]. Thus, preventing second messenger signaling specifically in lipid rafts induces a phenotype matching the alterations of the retinal projections in animals lacking a subset of ephrin-A receptors[47].

The chiasm of animals expressing the lipid raft-targeted or -excluded variant of SponGee or SpiCee were imaged at E18.5 using

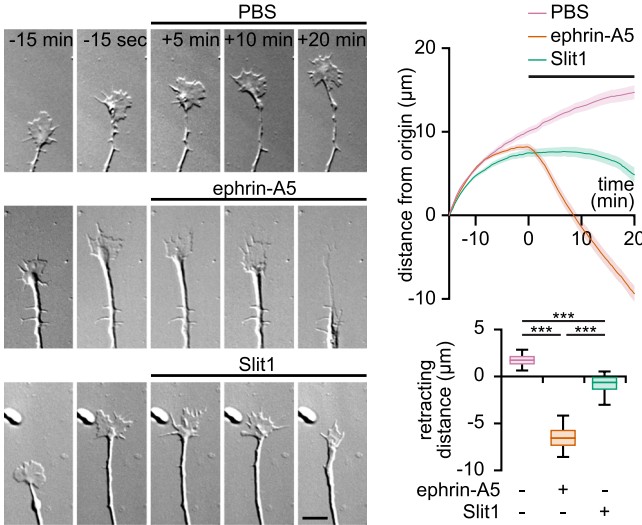

**Fig. 7 | Ephrin-A5 and Slit1 induce distinct morphological changes of axonal growth cones in vitro.** The growth of axons exposed to PBS is not affected (top row). Ephrin-A5 induces a growth cone collapse followed by a prompt retraction (middle row). Axons exposed to Slit1 exhibit a collapse of the growth cones but in contrast to axons encountering ephrin-A5, do not retract within the 20 min recorded (bottom row). Traces: mean ± s.e.m. Box-and-whisker plot elements: median, upper and lower quartiles, 10th and 90th percentiles. *** $P < 0.001$; Kruskal–Wallis test followed by Dunn's post-hoc test. Source data, number of replicates and $P$ values are provided as a Source Data file.

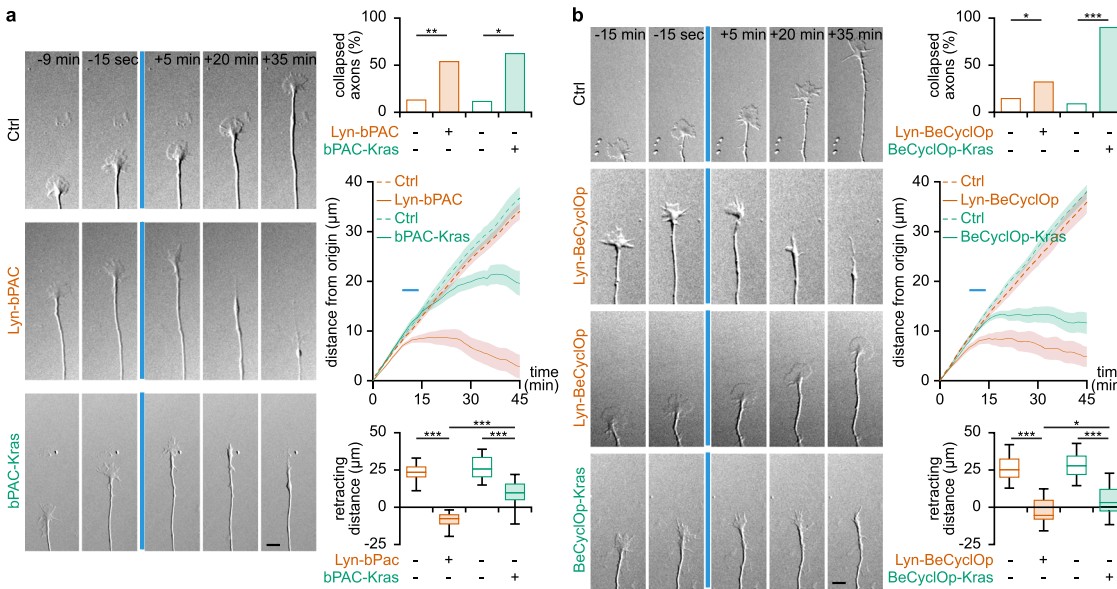

**Fig. 8 | Imposing lipid raft-restricted and -excluded cyclic nucleotides modulation mimics the axon behavior induced by ephrin-A5 and slit1, respectively. a** Lyn-bPAC-expressing growth cones retract when exposed to successive pulse of blue light whereas control axons are insensitive to this light stimulation. By contrast, bPAC-Kras-expressing growth cones collapse but do not retract. The blue line denotes the time of light exposure both in the image sequences and in the traces. **b** When exposed to blue light, Lyn-BeCyclOp induces the collapse of a limited fraction of growth cone. Light activation of BeCyclop-Kras also leads to growth

cone collapse, but in a large majority of axons. Lyn-BeCyclOp-expressing collapsing growth cones retract more than the axons of BeCyclOp-Kras-electroporated neurons. **a,b** Traces: mean ± s.e.m. Box-and-whisker plot elements: median, upper and lower quartiles, 10th and 90th percentiles. * $P < 0.05$, ** $P < 0.01$, *** $P < 0.001$. Top graphs $\chi^2$ test followed by $\chi^2$ post-hoc tests; bottom graphs, One way ANOVA followed by Dunnett post-hoc test. Source data, number of replicates and $P$ values are provided as a Source Data file.

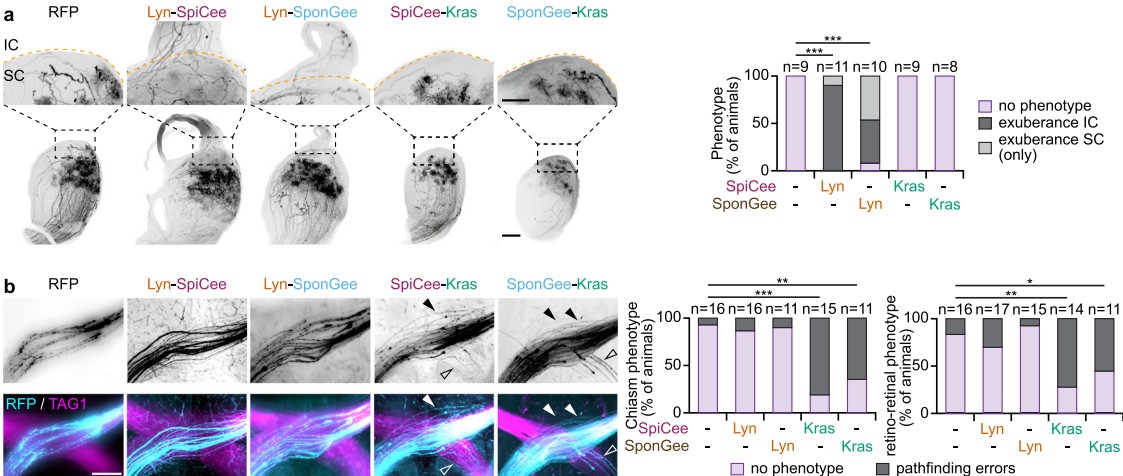

**Fig. 9 | Lipid raft-targeted and -excluded scavenging lead to misguided retinal axons in the SC and at the optic chiasm, respectively. a** Lyn-SpiCee- and Lyn-SponGee-expression lead to overshooting axons in the inferior colliculus at P15, by contrast to mRFP-, SpiCee-Kras- and SponGee-Kras-expression. The orange dashed line highlights the position of the posterior end of the superior colliculus (SC). The inferior colliculus (IC) is above this line. The top row images are magnifications of the regions of the bottom row images indicated by the black dashed squares. Scale bars: top row, 250 μm; bottom row, 500 μm. **b** SpiCee-Kras- and SponGee-Kras-expressing axons (cyan) exit the optic chiasm labeled with TAG1 (magenta), by contrast to the axons of mRFP-, Lyn-SpiCee and Lyn-SponGee-electroporated RGCs. An excess of retino-retinal axons is also detected in SpiCee-Kras- and SponGee-Kras-electroporated animals, as compared to mRFP-, Lyn-SpiCee- and Lyn-SponGee-electroporated RGCs. The top row highlights the mRFP channels in which electroporated axons are seen. Closed arrowheads, axons exiting the optic chiasm; open arrowheads, retino-retinal axons. Scale bar, 200 μm. **a,b** ** $P < 0.01$; *** $P < 0.001$; $χ^2$ test followed by $χ^2$ post-hoc tests. Source data are provided as a Source data, number of replicates and P values are provided as a Source Data file.

light sheet microscopy after whole brain clearing. To visualize the position of the unaffected chiasm, a TAG1 immunostaining was performed, enabling to visualize both the axons of electroporated and non-electroporated RGCs. mRFP-, Lyn-SpiCee- and Lyn-SponGee-electroporated axons follow the TAG1-labeled tract, whereas SpiCee-Kras and SponGee-Kras expression led to axons defasciculating and growing at a distance of the chiasm before joining the rest of the retinal axons in the optic tract (Fig. 9b). In addition, SpiCee-Kras- and Spon-Gee-Kras- but not Lyn-SpiCee- and Lyn-SponGee-electroporated animals exhibit an increased number of axons growing in the contralateral optic nerve as compared to mRFP-electroporated controls (Fig. 9b). The expression of cAMP Sponge-Kras but not Lyn-cAMP Sponge leads to similar observations (Supplementary Fig. 5). These phenotypes are reminiscent of the Slit1/2 double knock-out animals that also exhibit misguided axons at the optic chiasm[31,32].

The environment of developing retinal axons contains other guidance molecules than Slits and ephrin-As and other molecular mechanisms contribute to axon pruning and pathfinding. Even though the use of cellular compartment-specific second messenger buffers provides some specificity for a subset of cAMP, cGMP and Ca²⁺ signaling pathways, Slit- and ephrin-A-independent mechanisms might contribute to the observed phenotypes. Similarly, other axon cues or adhesion molecules that contribute to retinal axon pathfinding might still be fully functional because unaffected by second messenger signaling restricted to the tested compartments, thus explaining that not all electroporated axons exhibit axon pathfinding defects.

## Discussion
### Physiological relevance of second messenger compartmentation
Although a variety of second messenger microdomains have been described, the link between a biochemically-identified compartment and a function for such local signals has been scarcely studied. Investigating the physiological relevance of local second messenger signaling has remained challenging until recently with the lack of tools enabling to manipulate cAMP, cGMP and Ca²⁺ in subcellular domains of known biochemical identity. The development of genetically-encoded

scavengers buffering cAMP, cGMP and Ca²⁺ has open a way to control the concentration of these signaling molecules with subcellular resolution[35,40,41]. Using this approach, we provide a comprehensive description of the subcellular compartmentation of cyclic nucleotides and Ca²⁺ signals induced in retinal growth cones by two repellent axon guidance molecules. Surprisingly, we found that ephrin-A5 and Slit1 downstream signaling involve second messenger signaling in distinct submembrane compartments: lipid rafts and the non-raft domain of the plasma membrane, respectively. The interactions between cAMP, cGMP and Ca²⁺ are also domain-specific. We confirmed that these differences in signaling correlate with distinct behaviors of axons exposed to either ephrin-A5 or Slit1 in vitro. In vivo, altering second messenger signaling in or outside lipid rafts induces distinct axon guidance phenotypes. Lipid raft-restricted signaling controls axon pathfinding in areas of the brain where ephrin-A5 is involved, whereas lipid raft-excluded second messenger modulation is required where Slit1 is critical for retinal axon guidance. Investigating other second messengers in developing axons might extend this concept of subcellular compartmentation to a distinct set of signaling molecules, including lipidic second messengers that regulate axon pathfinding[48,49]. A similar approach has been used to demonstrate that cAMP and cGMP signaling restricted to the primary cilium or to the closely apposed centrosome regulate distinct features of cortical interneuron migration[50]. Overall, these studies demonstrate that the restriction of second messengers in micro or nanodomains is functionally relevant and is critical for the development of the nervous system. The approach used is adaptable to other systems to enlarge the investigation of the physiological processes requiring second messenger signals restricted to subcellular compartments.

How ephrin-A/EphA and Slit/Robo pathways modulate second messenger signaling is still unclear. The spatial specificity of each pathway might be related to the subcellular localization of the receptors, although the subcellular location of EphA and Robos is unclear. In lipid rafts, the ephrin-A-induced cAMP reduction is placed upstream the cGMP and Ca²⁺ changes, suggesting that a modulation of cAMP is sufficient to induce the other changes in second messenger signaling. Although the molecular link has not been identified, EphA activation

might lead to either a reduction in the resting activity of lipid raft-targeted adenylyl cyclases or the increased activity of phosphodiesterases (PDEs), the cAMP degrading enzymes that are also important to control the subcellular location of this second messenger. The downstream changes in cGMP might then rely on a cAMP-activated PDE that hydrolyses cGMP such as PDE5[51,52]. The observation that reduced cAMP concentration leads to an elevation of $Ca^{2+}$ transient frequency is more surprising since an elevated cAMP concentration has often been correlated with increased $Ca^{2+}$ channel activity. However, in the case of rhythmic behavior such as the one observed for $Ca^{2+}$ transients, cAMP enhances the activity of TREK potassium channels that favors refractory periods preventing activity in starburst amacrine cells[53]. Since the retraction response of retinal axons exposed to ephrin-A5 requires electrical activity[54], a reduction in cAMP might prevent TREK channels activation and release the possibility of voltage-dependent $Ca^{2+}$ channel opening, thus leading to a rhythmic $Ca^{2+}$ transient activity. Adjusting the molecules involved, a similar interaction network based on reciprocal inhibition between cAMP and cGMP and the control of the rhythmic $Ca^{2+}$ activity can be drawn outside lipid rafts downstream of the Slit/Robo pathway.

### Integration of multiple axon guidance cues occurs downstream of second messengers

The idea that second messenger signaling restricted to micro or nanodomains might activate distinct effectors downstream of axon guidance molecules has been formulated in particular to differentiate axon attractants and repellents[55]. However, the model accepted in most cases is that second messengers are placed in such a position in the signaling pathway of axon guidance molecules that they play the role of integrators. They might thus enable a switch between attraction and repulsion in a growth cone exposed to a mix of several guidance molecules. For instance, inverting the cAMP:cGMP ratio is sufficient to convert Netrin-1-induced attraction of spinal axons into repulsion[27]. The overall model to explain how $Ca^{2+}$ can regulate both attractive and repulsive behaviors of the growth cone is based on codes relying on signal amplitude of the response and the slope of the calcium gradient across the growth cone[55]. Our observations rather demonstrate that different axon guidance molecules induce second messenger signals in distinct compartments, thus placing cAMP, cGMP and $Ca^{2+}$ above a potential molecular integrator. Such an integrator should be in a position to combine second messenger signaling from different compartments of the growth cone. Since signaling pathways of some guidance molecules interact in a synergistic way[56], it is also conceivable that second messenger signals restricted to distinct cellular domains influence each other and contribute the integration of multiple cues, thus influencing the behavior of developing axons exposed to combinations of axon guidance molecules.

Strikingly, different second messenger signals regulate the behavior of developing axons facing distinct axon repellents, thus modulating subtle changes in growing axons. Lipid raft-excluded second messenger variations regulate the response to Slit1, an axon guidance cue that controls the path of growing retinal axons still at a distance from their target[31,32]. This axon repellent thus does not terminate the growth of the axon and enable its further extension in the optic tract. This is in line with the observation that Slit1 does not induce an immediate retraction in vitro. By contrast, ephrin-A5 is involved in shaping retinal projections in areas of the brain where axons form their terminal arbors and thus require a stop signal preventing further extension[33,47]. This stop signal involves second messenger signaling in lipid rafts. Consistently, ephrin-A5 induces the retraction of axons within minutes after their stimulation by this axon guidance molecule in vitro. These observations highlight that within the family of axon repellents, distinct axonal behaviors are regulated by cAMP, cGMP and $Ca^{2+}$ signals confined in different subcellular compartments, thus placing these second messengers upstream of a final process that integrates the diversity of guidance stimuli present in the environment of developing axons.

### Expanding the repertoire of available second messenger codes

The wide range of signaling pathways regulated by cyclic nucleotides and $Ca^{2+}$ requires a mechanism leading second messenger signals to achieve specificity for their downstream effector without interfering with the other cellular processes that these signaling molecules regulate. The subcellular compartmentation of cAMP, cGMP and $Ca^{2+}$ contributes to such specificity. For instance, subcellular-restricted signals have been identified in the filopodia of retinal growth cones, in T-tubule of cardiomyocytes, in lipid rafts or in the primary cilium[5,9,10,57]. Here, we provide evidence that in addition to subcellular compartmentation of second messengers, the crosstalk between cAMP, cGMP and $Ca^{2+}$ is highly controlled at a subcellular scale, with distinct interactions in the vicinity of lipid rafts and further away from this membrane compartment. Similarly cAMP and cGMP buffering in the primary cilium affects the polarity of migrating cortical interneurons in an opposite manner, whereas preventing the modulation of cyclic nucleotides at the centrosome leads to the same alteration of nucleokinesis, thus suggesting distinct co-regulations of these signaling molecules in these two compartments[50]. This is in line with the previously reported interactions between these signaling molecules, which are not conserved across cell types, suggesting multiple molecular controls of second messenger concentrations. Whereas $Ca^{2+}$ drives the drop of cAMP concentration in insulin-secreting MIN6 β-cells exhibiting combined cAMP and $Ca^{2+}$ oscillations, it is placed upstream of an elevation in cAMP in HEK cells with similar second messenger oscillations or in developing neurons[19–21]. However, these interactions were identified using cell-wide imaging and pharmacological manipulations, i.e., without subcellular resolution. Using such approaches, the detected crosstalk is thus likely to reflect the overall dominant pathway in each cell type, but might not report the diversity of second messenger interplays in distinct compartments within the same cell.

Achieving specificity by subcellular second messenger signals is limited by the number of available cellular domains that might not be sufficient to control the myriad of cAMP-, cGMP- and $Ca^{2+}$-dependent signaling pathways. A diversity of crosstalks activated by distinct stimuli within the same cellular compartment might be a mechanism enabling improved specificity. Within a subcellular domain, it is conceivable that the combination of second messenger modulations and their crosstalk might differ downstream of distinct receptors or signaling pathways, thus expanding the range of available second messenger codes when merging the diversity of subcellular domains with different signal combinations or crosstalks. This enlarged set of signals would enable to specifically activate a wider range of downstream effectors.

## Methods

### Animals

Pregnant C57BL6/J and RjOrl:SWISS mice were purchased from Janvier Labs. All animal procedures were performed in accordance with institutional guidelines and approved by the local ethics committee (C2EA-05: Comité d'éthique en expérimentation animale Charles Darwin; protocol APAFIS#22331-2019100814127972v4). Animals were housed on a 12 h light/12 h dark cycle in temperature- and humidity-controlled environment (19–23 °C, 45–60%). Embryos from dated matings (developmental stage stated in each section describing individual experiments) were not sexed during the experiments and the female over male ratio is expected to be close to 1.

### Retinal explants

Retinas of E14.5 mouse embryos were electroporated with using two poring pulses (square wave, 175 V, 5 ms duration, with 50 ms interval)

followed by four transfer pulses (40 V, 50 ms and 950 ms interpulse) with a Nepa21 Super Electroporator (NepaGene). This electroporation procedure was used for the following plasmids: mRFP, Lyn-SpiCee, mut-Lyn-SpiCee, SpiCee, SpiCee-Kras, mut-SpiCee-Kras, SponGee, Lyn-SponGee, mut-Lyn-SponGee, SponGee-Kras, mut-SponGee-Kras, cAMP Sponge, Lyn-Aspx, Aspx-Kras, mut-Aspx-kras, Lyn-H147, H147-Kras, Twitch2B, Lyn-Twitch2b, Twitch2b-Kras, $^{T}$HPDE5$^{VV}$, Lyn-RflincA, RflincA-Kras, Lyn-deltaflincG, δFlincG-Kras, Lyn-BeCyclOp, BeCyclOp-Kras, Lyn-bPAC, bPAC-Kras (2 μg μL$^{-1}$ for single plasmid electroporation, 1 μg μL$^{-1}$ for each plasmid for co-electroporation except for Lyn-BeCyclOp, BeCyclOp-Kras electroporated at 0.5 μg μL$^{-1}$). Retinas were dissected and kept 24 h in culture medium (DMEM-F12 supplemented with 1 mM glutamine (Sigma Aldrich), 1% penicillin/streptomycin (Sigma Aldrich), 0.001% BSA (Sigma Aldrich) and 0.07% glucose), in a humidified incubator at 37 °C and 5% CO2. The following day, the retinas were cut into 200 μm squares with a Tissue-Chopper (McIlwan) and explants were plated on glass coverslips or ibidi 35 mm glass bottom dish coated with 100 μg mL$^{-1}$ poly-lysine and 20 μg mL$^{-1}$ Laminin (Sigma Aldrich). Cells were cultured for 24 h in culture medium supplemented with 0.5% (w/v) methyl cellulose and B-27 (1/50, Life technologies).

## Molecular biology

Plasmids generated in this study (Lyn-Twitch2b, Twitch2b-Kras, Lyn-mut SponGee, mut SponGee- Kras, Lyn-mut SpiCee, mut SpiCee-Kras, Lyn-RFlincA, RFlincA-Kras, Lyn-BeCyclOp, BeCyclop-Kras, Lyn-δFlincG, δFlincG-Kras) are available upon request from the corresponding author. The other plasmids used have been deposited to Addgene under the following name and catalog number: SpiCee, #140836; Lyn-SpiCee, #140837; SpiCee-Kras, #140838; SponGee, #134775; Lyn-SponGee, #134776; SponGee-Kras, #134777; pCX-$^{T}$hPDE5$^{VV}$, #134778.

Mut SponGee carries the E247G and E299G mutations in the cGMP binding sites. The Lyn-mut SponGee insert was generated by gene synthesis (Thermofisher) and was subcloned into an empty pCX vector (containing a CAG promoter) between the Acc65I and NotI restriction sites. The Lyn targeting sequence flanked by two NheI sites was removed from Lyn-mut SponGee using NheI. Mut SponGee was then inserted into a mRFP-Kras-containing pCX plasmid between the AgeI and BsrGI restriction sites.

Mut SpiCee carries the following mutations: D52A, E63Q, D91A, E102Q, D135A, E146Q, D171A and E182Q. Mut SpiCee was subcloned into a Lyn-SpiCee vector between the BbvCI and BsrGI restriction sites, thus replacing the SpiCee sequence and generating the Lyn-mut SpiCee construct carrying a CAG promoter. Mut SpiCee was also subcloned into a SpiCee-Kras vector between the AgeI and BsrGI restriction sites, thus replacing the SpiCee sequence and generating the mut SpiCee-Kras construct under the control of a CAG promoter.

A plasmid containing a CAG promoter followed by a Lyn targeting sequence in frame with the Twitch2b sequence, a stop codon and the Kras sequence was generated using the In-Fusion kit (Ozyme) and oligonucleotides for the Lyn and Kras sequences. The Lyn sequence and the stop codon preceding the Kras sequence were excised using AgeI and AflII, respectively, to obtain the Twitch2b-Kras construct, whereas the Kras sequence was removed using NheI to obtain the Lyn-Twitch2b-encoding plasmid.

RFlincA was subcloned into a plasmid containing a CAG promotor using KpnI and XhoI. The pair of Lyn sequences was PCR amplified with the addition of a linker (CTCGAGGATCCA) and ligated into the plasmid using a Roche ligation kit to obtain Lyn-RFlincA.

RFlincA was subcloned into a plasmid containing a CAG promotor using AgeI and KpnI. The Kras sequence was PCR amplified with the addition of a linker (TGTACA) and ligated into the plasmid following the NEBuilder provider instructions to obtain RFlincA-Kras.

BeCyclOp was subcloned into a plasmid containing a CAG promotor using KpnI and BamHI. The pair of Lyn sequence was PCR amplified with the addition of a linker (GGATCCGCCACC) and ligated into the plasmid using a Roche ligation kit to obtain Lyn-BeCyclOp.

BeCyclOp was subcloned into a plasmid containing a CAG promotor using NheI and SacI. The Kras sequence was PCR amplified with the addition of a linker (GAGCTC) and ligated into the plasmid following the NEBuilder provider instructions to obtain BeCyclOp-Kras.

δFlincG was subcloned into a plasmid containing a CAG promotor using AgeI and KpnI. δFlincG and the pair of Lyn sequences were PCR amplified with the addition of a linker (GGTACCGTGGCA) and ligated into the plasmid following the NEBuilder provider instructions to obtain δFlincG-Kras.

δFlincG was subcloned into a plasmid containing a CAG promotor using NheI and SacI. δFlincG and the Kras sequence were PCR amplified with the addition of a linker (GGATCCGAGCTC) and ligated into the plasmid following the NEBuilder provider instructions to obtain δFlincG-Kras.

All plasmids were sequenced to verify the success of the cloning strategies

## Membrane fractionation by detergent-free method

Electroporated retinas were pelleted (195 g for 5 min at 4 °C) and resuspended in 1.34 mL of 0.5 M sodium carbonate, pH 11.5, with protease inhibitor cocktail and phosphatase inhibitor cocktails 1, 2 and 3 (Sigma-Aldrich). The homogenate was sheared through a 26-gauge needle and sonicated three times for 20 s bursts. The homogenate was adjusted to 40% (w/v) sucrose by adding 2.06 mL of 60% (w/v) sucrose in MBS (25 mM MES, pH 6.4, 150 mM NaCl, and 250 mM sodium carbonate), placed under a 5–30% (w/v) discontinuous sucrose gradient, and centrifuged at 34,000 rpm (142 × g, calculated in the middle of the centrifugation tube) for 15–18 h at 4 °C in a Beckman SW 41Ti rotor. Nine fractions (1.24 mL each) were harvested from the top of the tube mixed with 9 volumes of MBS, and centrifuged at 40,000 rpm for 1 h at 4 °C (Beckman SW-41Ti rotor, 197 × g, calculated in the middle of the centrifugation tube). Supernatants were discarded, and membrane pellets were resuspended in 100 μL of 1% (w/v) SDS.

For immunoblotting, samples were separated on a precast gel (4–15% Mini- Protean TGX Tris-Glycine-buffer SDS PAGE, Biorad) and transferred onto 0.2 μm Trans-Blot Turbo nitrocellulose membranes (Biorad). Membranes were blocked for 1 h at room temperature in 1xTBS (10 mM Tris pH 8.0, 150 mM NaCl) supplemented with 5% (w/v) dried skim milk powder. Primary antibody incubation was carried out overnight at 4 °C, with the following antibodies: rabbit anti-GFP (1/200; A11122; Life Technologies), rabbit anti-β-Adaptin (1/200; sc–10762; Santa Cruz; lot # E1304) and rabbit anti-Caveolin (1/500; 610060; BD Transduction Laboratories; lot # GR256941–5). All primary antibodies have been previously validated for this assay[9]. A goat anti-rabbit-HRP-coupled secondary antibody was used for detection (Jackson ImmunoResearch, West Grove, PA). After antibody incubations, membranes were extensively washed in TBS T (TBS containing 2.5% (v/v) Tween-20). Western blots were visualized using the enhanced chemiluminescence method (ECL prime Western Blotting detection reagent, Amersham).

## In utero retinal electroporation

In utero electroporation was performed like previously described[58,59]. In brief, timed-pregnant mice (Janvier Labs) were delivered to the animal facility a week prior to the surgery in order to allow a minimum of 5 days adaptation. C57BL/6NRj pregnant mice were anesthetized with an intraperitoneal injection of a Xylazine/Ketamine mix (10 mg kg$^{-1}$ and 100 mg kg$^{-1}$, respectively) and a subcutaneous injection of buprenorphine (0.0125 mg kg$^{-1}$) was made pre-surgery for analgesia. Midline laparotomy was performed, exposing uterine horns and allowing visualization of embryos. The left eye of E14.5 embryos was injected with 2 μg μL$^{-1}$ of DNA using an elongated glass capillary (Harvard apparatus) with different plasmid solutions. The success of

DNA injection was assessed using 0.07% fast green supplemented to the DNA solution. The eye was then electroporated with 5 pulses of 45 V during 50 ms every 950 ms (Nepagene electroporator). To target the central part of the retina, the positive electrode (CUY650P5, Sonidel) was placed on the side of the injected eye. Following surgery, the incision site was sutured (4-0, Ethicon), and mice were allowed to give birth. To increase the survival of the electroporated pups, a Swiss mouse was housed together with the mice that underwent surgery to favor the care of the pups. The Swiss mouse, mated a day earlier than the C57BL/6NRj mice, gave birth one day earlier. At P0, only 2 Swiss pups were left in the cage so that the electroporated pups were adopted by the Swiss mouse.

### Collapse assay
Retinal explants were treated with 200 ng mL$^{-1}$ rmSlit-1 or 500 ng mL$^{-1}$ rmEphrinA5 (R&D Systems) diluted in warm culture medium for 20 min before fixation with 4% (w/v) PFA in Sucrose 4% (w/v) for 30 min.

### Immunostaining following collapse assay
Retinal explants were permeabilized and blocked with 0.25% (v/v) Triton and 3% (w/v) BSA in PBS, then immunized against DsRed (1/1000, Takara Bio, lot #CDSO0219101) followed by a secondary antibody coupled to AlexaFluor 594 (1/500, Thermo Fisher Scientific) and βIII-tubulin (1/1000, Biolegend, lot #B249869) followed by a secondary antibody coupled to AlexaFluor 488 (1/500, Thermo Fisher Scientific). Antibodies were diluted in PBS supplemented with 0.1% (v/v) Triton and 1% (w/v) BSA.

### Whole mount immunostaining and tissue clearing
P15 mice were deeply anesthetized with a mix of Xylazine/Ketamine (20 mg kg$^{-1}$ and 200 mg kg$^{-1}$ respectively), perfused transcardially with 4% PFA in 0.12 M phosphate buffer. Retinas and brains were dissected out and post-fixed 12 h in 4% PFA. Retinas (oriented with an incision on the ventral part) and were mounted in Mowiol. To validate area of electroporation, retinas were imaged under a 2.5X objective using an epi-fluorescence microscope (Leica DMI6000B). E18.5 embryos were harvested and the heads were fixed 3 h with 4% PFA. The skulls were dissected out to harvest the brain and post-fixed 12 h in 4% PFA.

The samples were prepared according to the iDISCO+ protocol adapted from ref. 60. The brains were then dehydrated in succeeding baths of methanol/PBS for 1.5 h each at RT (50% MeOH, 80% MeOH, 100% MeOH). The samples were then transferred overnight in a depigmentation solution of methanol containing 6% H$_2$O$_2$ (VWR, 216763) at 4 °C. E18.5 samples were then depigmented for 3 additional days in a solution of methanol containing 10% H$_2$O$_2$ at 4 °C.

The samples were rehydrated in succeeding bath of methanol/PBS for 1.5 h each at room temperature (100% MeOH X2, 80% MeOH, and 50% MeOH, PBS) and kept in PBS at 4 °C before immunostaining.

The brains were permeabilized in the blocking solution (0.5% Triton-X100, 0.2% gelatin, 1X PBS, 0.1 g/L thimerosal) for 2 days for P15 brains and 24 h for E18.5 brains at room temperature on agitation. For immunostaining, the samples were incubated with the primary antibodies against TAG1 (1/500, R&D Systems, lot #CDSO0219101) and or DsRed (1/1000, Takara Bio, lot #2103116) in a solution containing 0.5% Triton-X100, 0.2% gelatin, 1XPBS, 0.1 g/L thimerosal, 10 mg/mL saponin. Incubation in the primary antibody solutions was performed for 2 weeks for P15 brains or 1 week for E18.5 brains at 37 °C under agitation. The samples were washed for 1 day. The samples were then immunized by secondary antibodies coupled to AlexaFluor 647 (1/500, Jackson Immunoresearch, lot #146920) and or Cy3 (1/500, Jackson Immunoresearch, lot #148687) diluted in the same solution as for the primary antibodies, passed through a 0.22 μm filter and incubated for 1 week for P15 brains and 2 days for E18.5 brains at 37 °C under agitation. The samples were then washed for 6 times during 1 day in PBS

supplemented with 0.2% gelatin and 0.5% Triton-X100, and 2 washes of 1X PBS prior to storing the samples in the dark at 4 °C until clearing.

For clearing, the samples were first transferred overnight in a solution containing 20% MeOH. The following day they were incubated in succeeding bath of Methanol/PBS for 1 h (40% MeOH, 60% MeOH, 80% MeOH, 100% MeOH X2). Sample were then placed overnight in a solution of 2/3 dichloromethane and 1/3 MeOH. The following day, they were incubated 30 min in a 100% dichloromethane bath prior being transferred in 100% benzyl ether (DBE) and stored until imaging.

### Light sheet microscopy
Images were acquired with an ultramicroscope I (LaVision BioTec, Miltenyi Biotec) coupled to a 2× objective (Olympus, MVPLAPO) with different magnifications (0.63×, 1×, 1.25×, 1.6×, 2×, 2.5×, 3.2×, 4× and 5×) or a 12× objective and with the ImspectorPro software (LaVision Biotec, Miltenyi Biotec). The light sheet was generated by a laser (wavelength 561 and 640 nm, coherent Sapphire Laser, LaVision BioTec, Miltenyi Biotec). Samples were imaged in DBE with a Zyla SCMOS camera (Andor, Oxford Instrument). Step size between each image was fixed at 1 μm (NA = 0.5, 150 ms time exposure).

### FRET imaging and analysis
Images were acquired with an inverted DMI6000B epi-fluorescence microscope (Leica) coupled to a 40× oil-immersion objective (N.A. 1.3) and Metamorph software (Molecular Devices). Retinal explants were superfused (0.3 mL min $^{-1}$) using a close chamber (FCS2, Bioptechs) and a syringe-pump (Aladdin, WPI), to avoid imaging artefacts generated by the pulses of peristaltic pumps. The superfusion medium was adapted from the culture medium: 1 mM CaCl$_2$, 0.3 mM MgCl$_2$, 0.5 mM Na$_2$HPO$_4$, 0.45 mM NaH$_2$PO$_4$, 0.4 mM MgSO$_4$, 4.25 mM KCl, 14 mM NaHCO$_3$, 120 mM NaCl, 0.0004% CuSO$_4$, 0.124 mM Fe(NO$_3$)$_3$, 1.5 mM FeSO$_4$, 1.5 mM thymidine, 0.51 mM lipoic acid, 1.5 mM ZnSO$_4$, 0.5 mM sodium pyruvate (all from Sigma), 1X MEM Amino Acids (Life Technologies), 1X non-essential amino acids (Life Technologies), 25 mM HEPES (Sigma), 0.5 mM putrescine (Sigma), 0.01% BSA (Sigma), 0.46% glucose (Sigma), 1 mM glutamine (Life Technologies), 2% penicillin streptomycin (Life Technologies). Vitamin B12 and riboflavin were omitted because of their auto-fluorescence. rmSlit-1 was used at 200 ng mL$^{-1}$ and rmEphrinA5 at 500 ng mL$^{-1}$ (R&D Systems). Spermine-NONOate was used at 50 μM, Forskolin (Sigma) at 10 μM and ionomycin (Invitrogen) was used at 5 μM. Images were acquired simultaneously for the CFP (483/32 nm) and YFP (542/27) channels every 20 s for cAMP or cGMP detection, or every 5 s for calcium detection, while cells were continuously superfused with the medium described above. Simultaneous CFP and YFP channel acquisition was achieved using a dual chip CCD camera ORCA-D2 (Hamamatsu). The wavelength used for CFP excitation was 436/20 nm.

Images were processed in ImageJ, corrected for background and bleedthrough from CFP into the YFP channel, and the CFP:YFP (H147) or the YFP:CFP ($^{T}$hPDE5$^{VV}$, Twitch2b) ratio was computed for each axon. The cAMP and cGMP data were analyzed using two parallel pipelines. The first analysis was used to generates the histograms of Figs. 1–5 and Supplementary Fig. 2, and for statistical analyses: The measured ratios were averaged over two 1 min-long time periods:] 9–10 min], i.e., immediately before the opening of the valve controlling the ephrin-A5 or Slit1 application, and [11–12 min[, i.e., providing a 1 min delay after the valve opening because of the dead volume of our perfusion system. The ratio was then normalized using the following formula: $\frac{\Delta R}{R_0} = \frac{\bar{R}_{[11-12min[} - \bar{R}_{]9-10min]}}{\bar{R}_{]9-10min]}}$ .

The second analysis pipeline for cyclic nucleotide FRET experiments was used to produce the illustration traces shown in Figs. 1–5 and Supplementary Fig. 2. The trace obtained for each axon was normalized to the R$_0$ described above (average of the measured ratio over the]9–10 min] period, i.e., immediately before the opening of the valve

controlling the ephrin-A5 or Slit1 application). For each experimental condition, traces of all axons were then averaged. Since sham experiments highlighted a drift of the ratio over time starting before the stimulation, traces were corrected for this drift to minimize the progressive elevation of the normalized ratio over the time period containing the 5 min immediately prior the stimulation. This correction is linear and uses the following formula: $\frac{\triangle R_{\text{corrected}}(t)}{R_0} = \frac{\triangle R_{\text{raw}}(t)}{R_0} - \alpha \cdot t$ where $t$ is the time, $\frac{\triangle R_{\text{corrected}}(t)}{R_0}$ the corrected normalized ratio, $\frac{\triangle R_{\text{raw}}(t)}{R_0}$ the normalized ratio before the drift correction, and $\alpha$ the slope of the linear correction. This method provides an easier visual interpretation of the trace but does not affect the statistics since statistical analyses are performed on the uncorrected dataset.

The Ca²⁺ FRET ratio was normalized to the ratio of the first image of the movie. No further computing was performed on the traces. The number of Ca²⁺ transients was automatically detected. All transients with an amplitude of 2.5 times the standard deviation were counted. A manual check was performed after this initial automatic quantification.

### DIC imaging and analysis (ephrin-A5 or Slit1 stimulation)
Images were acquired with an inverted Eclipse Ti2 (Nikon) coupled to a 40× oil-immersion objective and Metamorph software (Molecular Devices). Retinal explants were imaged in culture medium without phenol red and supplemented with 2 μM HEPES at 37 °C. rmSlit-1 was used at 200 ng mL⁻¹ and rmEphrinA5 at 500 ng mL⁻¹ (R&D Systems). Images were acquired in DIC every 15 s by using a Prime95B camera (Photometrics). Images were processed in ImageJ, growth length was measured with the Manual tracking plugin.

### DIC imaging and analysis (optogenetic stimulation)
Images were acquired with an inverted Eclipse Ti2 (Nikon) coupled to a 40× oil-immersion objective and Metamorph software (Molecular Devices). Retinal explants were imaged in culture medium without phenol red and supplemented with 2 μM HEPES at 37 °C. Images were acquired in DIC every 1 min by using a Prime95B camera (Photometrics). Retinas were co-electroporated ex vivo with Lyn-δFlincG, δFlincG-Kras, Lyn-RflincA or RflincA-Kras and either Lyn-BeCyclOp, BeCyclOp-Kras, Lyn-bPAC or bPAC-Kras. At the beginning of each experiment, an image was acquired in the RFP channel (561 nm excitation) to identify bPAC or Becyclop-expressing axons. Both constructs are fused to mRFP. After monitoring growth for 10 min using a low intensity of transmitted light to minimize bPAC or BeCyclOp stimulation, axons were stimulated by five laser flashes (445 and 491 nm, 100 ms of light exposure per flash, 1 min between flashes). After stimulation, axons were monitored for an additional 25 min, using the low intensity transmitted light. Images were processed in ImageJ, growth length was measured with the Manual tracking plugin of ImageJ and the number of axons responding to light (either collapsing or retracting) was quantified.

### Optogenetic tools validation and analysis
Images were acquired using an inverted Eclipse Ti2 (Nikon) coupled to a 40× oil-immersion objective and Metamorph software (Molecular Devices). Retinal explants were imaged in culture medium without phenol red and supplemented with 2 μM HEPES at 37 °C. Retinas were co-electroporated ex vivo with Lyn-δFlincG, δFlincG-Kras, Lyn-RflincA or RflincA-Kras and Lyn-BeCylcOp, BeCyclOp-Kras, Lyn-bPAC, bPAC-Kras, respectively. Images were acquired at 561 nm every minute for cAMP detection and or 491 nm every other minute for cGMP detection by using a Prime95B camera (Photometrics). After a first imaging period of 5 min, axons were stimulated by shining five flashes of light (wavelength 445 and 491 nm, 100 ms light exposure per flash, 1 min between flashes). After stimulation, axons were monitored for an additional 15 min. Images were processed in ImageJ, corrected for background and the fluorescence intensity values of the RFP channel for cAMP or GFP channel for cGMP was computed and normalized by

the fluorescence measured at the beginning of the recording to produce the traces shown in Supplementary Fig. 4.

The images were analyzed using the following pipeline for statistical analysis. The fluorescence intensity was measured and averaged over two time periods: [0–4 min] for cGMP and [3–5 min] for cAMP, i.e., immediately before the light stimulation, and [6–10 min] for cGMP and ]5–7 min] for cAMP, i.e., immediately after the first flash of light. The change in the fluorescence intensity was then normalized using the following formula: $\frac{\Delta F}{F_0} = \frac{\bar{F}_{[6-10\text{min}]} - \bar{F}_{[0-4\text{min}]}}{\bar{F}_{[0-4\text{min}]}}$ for cGMP and: $\frac{\Delta F}{F_0} = \frac{\bar{F}_{[5-7\text{min}]} - \bar{F}_{[3-5\text{min}]}}{\bar{F}_{[3-5\text{min}]}}$ for cAMP.

### Statistical analysis
No data were excluded from the analysis, except for FRET imaging for which axons lacking a NO- ionomycin- or Forskolin-induced change in FRET reflecting an elevation of cGMP, Ca²⁺ or cGMP, respectively, were excluded from the analysis. No sample size calculation was performed. Sample size was considered sufficient after at least three independent experiments, leading to $n \geq 3$ since several animals, coverslips, or biochemical assays were often analyzed for the same experimental condition. Animals or cultures were equivalent and not distinguishable before treatment, *de facto* randomizing the sample without the need of a formal randomization process. Photomicrographs were often easily traceable by eye to its experimental condition, making blind analysis of the data difficult to achieve. When careful blinding was performed, experiments reproduced the results obtained in non-blinded experiments with identical experimental conditions. Image calculation and analysis were performed using ImageJ.

Statistical tests were calculated using GraphPad Prism (GraphPad Software Inc.). The number of replicates for all the data shown in Figs. 1–9 and Supplementary Figs. 1–5 is provided in the Source Data file.

### Reporting summary
Further information on research design is available in the Nature Portfolio Reporting Summary linked to this article.

## Data availability
The data generated in this study are provided in the Source Data file. Source data are provided with this paper.

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

## Acknowledgements

We thank Dr. O. Griesbeck (Twitch2b), Dr. G. Nagel (BeCyclOp, bPAC), Dr. K. Horikawa (RFlincA), Dr. A. Hofer (cAMP Sponge) and Dr. K. Jalink (H147) for the gift of the constructs that they developed. We are grateful to Christine Métin, Melody Atkins, Camille Michaud and the members of our lab for thoughtful discussion and helpful critical reading of the paper, and to the members of the animal and imaging facilities of Institut de la Vision. This work was supported, by grants from ANR (ANR-18-CE16-0017), Fondation pour la Recherche Médicale (EQU202003010158) and Fondation Voir et Entendre to X.N. This work was performed in the framework of LABEX LIFESENSES (ANR-10-LABX-65) and of IHU FOReSIGHT (ANR-18-IAHU-0001) supported by French state funds managed by the Agence Nationale de la Recherche within the Investissements d'Avenir program. S.B. was supported by fellowships from the ED3C doctoral program (Sorbonne Université) and Fondation pour la Recherche Médicale (FDT202106013022). J.B. was supported by a fellowship from Fondation de France (00099274) and Fondation pour la Recherche Médicale (FDT202204014862).

## Author contributions

Conceptualization, O.R., X.N.; Methodology, S.B., Y.Z., O.R., X.N.; Validation, S.B., O.R., X.N.; Formal Analysis, S.B., Y.Z., C.G.B., J.B., O.R., X.N.; Investigation, S.B., Y.Z., F.R., C.G.B., J.B., M.B., S.C., S.U., J.V.; Writing—Original Draft, S.B., X.N.; Writing—Review & Editing, S.B., Y.Z., F.R., C.G.B., J.B., M.B., S.C., O.R., X.N.; Visualization, S.B., X.N.; Supervision, O.R., X.N.; Project Administration, X.N.; Funding acquisition, X.N.

## Competing interests

O.R. and X.N. hold patents describing SpiCee and SponGee. The remaining authors declare no competing interests.
