## [Peer review file · Nature Communications]

REVIEWER COMMENTS

Reviewer #1 (Remarks to the Author):

Subcellular second messenger networks drive distinct repellent-induced axon behaviors

Baudet et al., received October 2022

In this paper, Baudet et al. investigate the role of the second messengers cAMP, cGMP and Ca²⁺ and how their dynamics are influenced by the presence of the guidance molecules ephrin-A5 and Slit1. The authors use specific genetic tools that allow them to unravel the subcellular compartmentation of these molecules in a clean and precise way. Moreover, they step further by analysing the effects of these molecules in the axon morphology both in vitro and in vivo. These experiments show the functional relevance in understanding the diverse role of the second messengers in micro and nanodomains.

Although this study has provided new and value data about the crosstalk of specific second messengers in the axonal behavior, some issues or questions need to be address.

- The authors use electroporation to transfer the sensors to the cells. Although they showed that the experiment is analysed in a considerable number of axons, it must be considered that each electroporation is different, as such, the cells would receive a distinct amount of plasmid and consequently sensor. Considering the sensitivity of these experiments, this could impact the results significantly. In some of the experiments, the replicates could be taken as enough, while others showed results from 3 different experiments. In any case, also the sample size calculation should be performed.
- The authors choose Ephrin-A5 and Slit, two axon repellent molecules. Do they consider approaching these experiments comparing to an attractant one? Why do they choose these ones in particular?
- This study is focus on three second messengers cAMP, cGMP and Ca²⁺. Do the authors have the tools to approach other second messengers?
- The authors use two repellents in this work, and they dissect their different influence in the second messengers. Have the authors tried to evaluate the effect of the presence of both Ephrin-A5 and Slit1? Which would be the expected consequences considering their effects on cAMP, cGMP and Ca²⁺?
- Figure 1a: the schemas presented in the figures are nice and clear. However, the Lyn targeting sequence and Kras targeting sequence should be presented in another colour or in a more visible shape so they can be seen better.
- Figure 1-4: Although the colouring of the figures very helpful to track all the different conditions the authors are addressing, some of them could be difficult to follow for a colourblind person and needs to be taken in account.
- Figure 2 a: A combination of the graphics resembling the format on Figure 1 is suggested, as it would be easier to compare among the conditions.
- Figure 3: The experimental condition of Ephrin-A5 + Lyn-cAMP and Lyn-SpiCee is missing.
- Figure 4: The experimental condition of Ephrin-A5 cAMP Sponge is missing.
- Figure 6: the authors mentioned the Slit1/2 DKO mutant. It could be interesting to approach a rescue experiment in this model considering what they have demonstrated in this paper.
- Figure 1b, Figure 2b, Figure 3bf: the design of the before and after vehicle/cue graph could be modified to be more visible, instead of faint empty circle, could be filled circle.
- In terms of references:
 - o some papers that could have relevance for discussing in this topic are not mentioned, as Kobayashi et al. (Scientific Reports, 2013).
 - o The reference 15 is a review, the authors should be refereeing to the original articles.
- In the methods, in the line 665 it is said that HEK293 cells are used in the head title, but HEK293T in the description of them.
- In the line 669, 784 and 803: biology and imaging should not have a capital letter.
- In the line 739: there are two / when should be only one.
- In the line 800: ImageJ (Fiji).

- In the line 823 it is said Figures 1 to 7, but the paper has 6 main figures. This should be corrected to Figures 1-6.

Reviewer #2 (Remarks to the Author):

In this manuscript, the authors have used FRET-based sensors for cAMP, cGMP and Ca²⁺, either untargeted, targeted to lipid rafts (Lyn-sensor) or targeted outside lipid rafts (sensor-Kras) to study intracellular changes in these signalling molecules in retinal ganglion neurons following stimulation with two different axon repellants, Ephrin-A5 and Slit1. They also combine these sensors with scavengers of these signalling molecules, cAMP sponge for cAMP, SponGee for cGMP and SpiCee for Ca²⁺, also untargeted, targeted to lipid rafts (Lyn-sponge) or targeted outside lipid rafts (sponge-Kras) to demonstrate the effect of preventing/reducing the changes in each signalling molecule in the respective compartments on signalling as well as axon retraction by Ephrin-A5 and Slit1 and also axon guidance in vivo. The combination of these molecular tools makes this an elegant study that demonstrates how different localised changes in intracellular signalling molecules can contribute to different cellular effects.

Comments:

1. The quantitative interpretation of the traces must represent a challenge, given the relatively low signal compared to the noise. However, there is no description in the manuscript about how the traces are interpreted, except that the data depict the amplitude (according to the figure legends). Please provide details about how the amplitude is measured. If relying on the amplitude after stimulation gives large variability, perhaps the authors could consider if measuring AUC (area under the curve) after agonist application would be a more robust measure of changes in FRET?
2. What is the basis for the mean \pm SEM in the traces as opposed to the data included in the box/whisker plots? Please write this clearly. E.g. are the traces based on several cells from the same slide, whereas the box/whisker plots are from all the cells from all the slides? Frequently the mean \pm SEM data in the traces do not seem to match well the data in the box/whisker plots, e.g., in figure 2a (cytosolic H147 sensor), where the Slit1-induced mean change in FRET as read from the traces indicates a \sim 5% decrease, whereas the box/whisker plot indicates a \sim 2% increase, so please clarify.
3. On the same note, in figure 2a (H147-Kras): The average traces shows first a decrease then an increase? Which value is reported as the amplitude? The maximal negative value? This is important, as the average in the box/whisker is 0, although the trace seems to indicate that there is a reduction in FRET (ie. a reduction in cAMP).
4. For some biosensors (for example epac (ancestor of H147 biosensor used in the current manuscript)), adding a targeting sequence decreases the maximal efficacy of the biosensor (DOI: 10.1038/ncomms15031), limiting the comparison across different biosensors. Does the Lyn- or Kras-H147 show altered efficacy compared to cytosolic H147 (from the average traces, the maximal response (Fsk) seems reduced for the Lyn-H147)? If so, would it be more appropriate to present FRET signal as a percent of maximal FRET obtained with Fsk in figure 2a?
5. The targeted twitch biosensors developed in the current manuscript would be valuable tools for the biosensor community. Related to the questions on the targeted H147 biosensors (above), does the targeting affect the maximal efficacy of the Ca²⁺ biosensors? How would this affect the measurements?
6. There are still some mechanistic aspects that are not clear and that should at least be discussed, e.g.: How does Ephrin-A5 and Slit1 reduce cAMP? How is reduced cAMP linked to reduced Ca²⁺?

Minor:

Some of the figures would be easier to read if the sensor used were indicated (as already done in some of the figures)

There is inconsistency in the spelling of SponGee vs. Spongee in the figures

Line 581: missing R in KRas

Line 787-789: subscript lacking in formulas

Line 794-795: please check final concentrations of spermine-NONOate and forskolin that cells were exposed to. 50mM and 10mM seem more like stock concentrations and could lead to off-target effects if cells are exposed to these concentrations.

HEK293 cells: described in M&M, not clear which experiments were performed in HEK293 cells.

Reviewer #3 (Remarks to the Author):

In this manuscript, Baudet et al. have investigated second messenger signaling in subcellular compartments, so-called raft and non-raft membrane microdomains, in retinal ganglion cell axons responding to the repulsive cues ephrin-A5 and Slit1. Using latest tools to monitor or scavenge cAMP, cGMP and Ca²⁺ in these subcellular compartments, the authors showed that ephrin-A5 and Slit1 exert their repulsive activities through distinct signaling networks in raft and non-raft microdomains, respectively. I think that their findings substantially elaborate our understanding of molecular signaling underlying axon guidance. Below are my comments that I believe will help improve the quality of this work.

1. The authors have stated, e.g., in the abstract, that ephrin-A5 and Slit1 activate subcellular-specific second messenger crosstalks, each signaling network controlling distinct axonal morphology changes in vitro and pathfinding decisions in vivo. However, it is unclear whether the observed difference in axonal responses in vitro (Fig. 5) depends on subcellular localization of second messenger crosstalks. Also, the in vivo data in Fig. 6 are difficult to interpret because the reagents used in these experiments can perturb second messengers that are unrelated to the downstream signals of ephrin-A5 and Slit1.

2. Although it is concluded in line 130 that Twitch2b-Kras is not targeted to lipid rafts (Supplementary Fig. 1), the immunoblots in this figure actually indicate the presence of Twitch2b-Kras in raft fractions, e.g., a substantial amount of Twitch2b-Kras in the fraction 4 colocalizing with caveolin.

3. The data in Figure 3 are consistent with the proposed models of signaling networks (d and h) but do not substantiate the validity of these models. The authors should tone down their description about these signaling networks.

4. How do ephrin-A5 and Slit1 elicit subcellular-specific second messenger signaling? Are receptors of ephrin-A5 and Slit1 localized to raft and non-raft domains of growth cone plasma membranes, respectively?

Reviewer #4 (Remarks to the Author):

The present manuscript addresses the question of how second messengers discriminate between different repellent axon guidance cues. The authors use genetically-encoded scavengers to define the subcellular compartmentation of cyclic nucleotides (cAMP and cGMP) and Ca²⁺ signals to define the response of retinal growth cones to Slit1 and ephrin-A5, two repellent axon guidance molecules that are known to influence these axons along their pathway up to the superior colliculus. They show that second messenger signaling for ephrin-A5 and Slit1 and their interaction occurs in two distinct submembrane compartments: lipid rafts and the non-raft domain of the plasma membrane. They further used in vitro and in vivo approaches to show that altering the subcellular distribution of the second messengers induces axon guidance defects. They conclude that cAMP, cGMP and Ca²⁺ do not really act as molecular integrators of axon guidance cues, as previously thought, but rather provide

distinct information in part encoded by different subcellular localization.

This is an interesting study that offers conceptual advances related to the mechanism that allow a growth to interpret different repellent guidance cues. The study is in general well conducted.

I have a few comments that may help improving the study:

1. Figure 1-3 are rather packed and difficult to follow. Can the authors display the items in a clear way? Can they also show a few examples of what are the actual images they use to obtain the different graphs? This will help the reader to understand better how the data were obtained.
2. All the graphs related to the frequency of Ca^{2+} transients show a variability within the same condition and several samples do not seem to have any response. What is the explanation for this variability?
3. The in vivo data show limited defects in few of the electroporated axons, why?
4. Slit1 controls intra-retinal axons guidance. Have the authors check if there is a retinal phenotype in the different conditions
5. The text is not always easy to follow and some sentences may benefit from some simplification

We are grateful to the reviewers for their positive and thoughtful assessment of our work.

Reviewer #1 (Remarks to the Author):

In this paper, Baudet et al. investigate the role of the second messengers cAMP, cGMP and Ca²⁺ and how their dynamics are influenced by the presence of the guidance molecules ephrin-A5 and Slit1. The authors use specific genetic tools that allow them to unravel the subcellular compartmentation of these molecules in a clean and precise way. Moreover, they step further by analysing the effects of these molecules in the axon morphology both in vitro and in vivo. These experiments show the functional relevance in understanding the diverse role of the second messengers in micro and nanodomains.

Although this study has provided new and value data about the crosstalk of specific second messengers in the axonal behavior, some issues or questions need to be address.

We thank the reviewer for her/his positive evaluation of our study.

- The authors use electroporation to transfer the sensors to the cells. Although they showed that the experiment is analysed in a considerable number of axons, it must be considered that each electroporation is different, as such, the cells would receive a distinct amount of plasmid and consequently sensor. Considering the sensitivity of these experiments, this could impact the results significantly. In some of the experiments, the replicates could be taken as enough, while others showed results from 3 different experiments. In any case, also the sample size calculation should be performed.

We agree with the reviewer that electroporation might introduce some variability since the level of expression differ between cells. We evaluated the impact of the level of expression on SpiCee and SponGee expression (together with the appropriate sensor) in two previous reports (Ros et al 2019; Ros et al. 2020). We did not notice any noticeable effect of the level of expression once it exceeds a minimal threshold.

As noted by the reviewer, the experiments have been performed on a considerable number of axons. We report both the number of independent experiments (*i.e.*, cell culture days) but also of distinct coverslips, meaning completely distinct recordings. The later often exceeds the number of independent experiments and might be considered as replicates.

Since sample size is calculated a priori and requires a wide range of assumptions about the distribution of the data, we think that such calculation is hardly doable in the case of the experimental design of our study. Post-hoc evaluation of the statistical power might provide a similar kind of information. We calculated the power for each relevant comparison. It is now available in Table S1.

- The authors choose Ephrin-A5 and Slit, two axon repellent molecules. Do they consider approaching these experiments comparing to an attractant one? Why do they choose these ones in particular?

Extending the ephrin-A5 and Slit1 data sets to an axon attractant is a very interesting suggestion although it would require an amount of work that is not compatible with the duration of the revision process. It has been approached in Nicol et al 2011 (cited in the manuscript), although only partially, in a different neuronal subtype (spinal neurons) and without the identification the biochemical identification of morphologically- defined growth cone compartments (filopodia vs growth center).

The guidance molecule studied was Netrin-1. Extrapolating from this study and combining it with ours, we may suggest that attractants induce changes in the cAMP concentration in the opposite direction (elevation for attractants vs reduction for repellents). In contrast, both attractants and repellents might require an elevation in the Ca^{2+} transient frequency. However, we feel that this extrapolation is not based on enough data to be included in our manuscript.

The rationale to choose ephrin-A5 and Slit1 was the following:

- (i) Both of them are required for the pathfinding of retinal axons
- (ii) Both of them repel retinal axons. This enables to evaluate whether molecules that induce similar, although slightly different (see Fig. 7), axon behavior share the same kind of second messenger signals. This has been suggested in the fields but our study invalidate this hypothesis. This point is mentioned in the discussion (paragraph starting line 385).

- This study is focus on three second messengers cAMP, cGMP and Ca^{2+} . Do the authors have the tools to approach other second messengers?

Once again, the point raised by the reviewer is very interesting. We unfortunately do not have a toolset ready for other second messengers and feel that this might be a great follow-up of the study described in the manuscript. We mentioned the possibility of such a future study in the discussion (line 357).

- The authors use two repellents in this work, and they dissect their different influence in the second messengers. Have the authors tried to evaluate the effect of the presence of both Ephrin-A5 and Slit1? Which would be the expected consequences considering their effects on cAMP, cGMP and Ca^{2+} ?

The question raised by the reviewer is of great interest since combinations of axon guidance molecules often induce synergistic rather than additive signaling pathways. It might be the focus of a future study. We feel however that it is outside the scope of the present manuscript that describes the interactions between second messenger within a subcellular compartment rather than a characterization of the interactions between compartments. We mentioned this point in the discussion (line 397).

- Figure 1a: the schemas presented in the figures are nice and clear. However, the Lyn targeting sequence and Kras targeting sequence should be presented in another colour or in a more visible shape so they can be seen better.

We thank the reviewer for this suggestion. We changed the color palette of the figures for a color-blind friendly color set (see another comment of this reviewer below) and to have an easier identification of the Lyn and Kras targeting sequences.

- Figure 1-4: Although the colouring of the figures very helpful to track all the different conditions the authors are addressing, some of them could be difficult to follow for a colourblind person and needs to be taken in account.

We thank the reviewer for this suggestion. We changed the color palette of all figures for a colorblind friendly color set.

- *Figure 2 a: A combination of the graphics resembling the format on Figure 1 is suggested, as it would be easier to compare among the conditions.*

We thank the reviewer for this suggestion. We added a schematic explaining the monitoring strategy for cAMP (Fig.2a). Since the schematics would be exactly the same as in Figure 1&2, for the following figures, we did not repeat the panels from Figure 3.

- *Figure 3: The experimental condition of Ephrin-A5 + Lyn-cAMP and Lyn-SpiCee is missing.*

- *Figure 4: The experimental condition of Ephrin-A5 cAMP Sponge is missing.*

We are not completely sure to identify the missing conditions mentioned by the reviewer. Nevertheless, some of these conditions have been added to Figure S2 (control of the specificity of the local second messenger buffers) and other are present in Averaimo et al. (Nature Commun, 2016), a publication from the same group using the exact same biological and experimental system. With the present manuscript and the mentioned published article, all combinations are now covered.

- *Figure 6: the authors mentioned the Slit1/2 DKO mutant. It could be interesting to approach a rescue experiment in this model considering what they have demonstrated in this paper.*

We agree with the reviewer that this experiment would be very interesting. However, the current available tools do not allow to impose specific spatiotemporal patterns of second messenger modulations in developing embryos. The closest toolset to approach this experimental suggestion would require optogenetic strategies in utero in the chiasm. Implanting an optic fiber in utero is out of reach of the available experimental approaches. Nevertheless we carried out a similar kind of experiment *in vitro* to mimic ephrin-A5- and Slit1-induced axonal behavior with optogenetic subcellular control of cyclic nucleotide concentration in and outside lipid rafts, respectively. These experiments are illustrated in Fig. 8.

- *Figure 1b, Figure 2b, Figure 3bf: the design of the before and after vehicle/cue graph could be modified to be more visible, instead of faint empty circle, could be filled circle.*

We adjusted the figures according to the reviewer suggestion.

- *In terms of references:*

o some papers that could have relevance for discussing in this topic are not mentioned, as Kobayashi et al. (Scientific Reports, 2013).

This reference has been included in the manuscript.

o The reference 15 is a review, the authors should be refereeing to the original articles.

Following the reviewer suggestion, we now provide an additional reference to exemplify the cyclic nucleotide-dependent regulation of phosphodiesterases (ref 16, not a review). However, we think that it is relevant to keep the review (ref 15) as a reference, since it summarizes the regulation of the large phosphodiesterase family. It provides a way to summarize the crosstalk between cAMP and cGMP without a likely too long explanation describing the regulations of all phosphodiesterases.

- In the methods, in the line 665 it is said that HEK293 cells are used in the head title, but HEK293T in the description of them.

Although there was a HEK293 cells section in the original manuscript, no experiments using these cells are described in the manuscript. We apologize for this mistake and removed this section in the revised manuscript.

- In the line 669, 784 and 803: biology and imaging should not have a capital letter.
- In the line 739: there are two / when should be only one.

We apologize for these typos. There are corrected in the revised manuscript.

- In the line 800: ImageJ (Fiji).

ImageJ was use instead of Fiji. Accordingly, we then did not add the mention to the Fiji bundle of ImageJ.

- In the line 823 it is said Figures 1 to 7, but the paper has 6 main figures. This should be corrected to Figures 1-6.

We apologize for this mistake. It is now corrected with the appropriate number of figures.

Reviewer #2 (Remarks to the Author):

In this manuscript, the authors have used FRET-based sensors for cAMP, cGMP and Ca²⁺, either untargeted, targeted to lipid rafts (Lyn-sensor) or targeted outside lipid rafts (sensor-Kras) to study intracellular changes in these signalling molecules in retinal ganglion neurons following stimulation with two different axon repellants, Ephrin-A5 and Slit1. They also combine these sensors with scavengers of these signalling molecules, cAMP sponge for cAMP, SponGee for cGMP and SpiCee for Ca²⁺, also untargeted, targeted to lipid rafts (Lyn-sponge) or targeted outside lipid rafts (sponge-Kras) to demonstrate the effect of preventing/reducing the changes in each signalling molecule in the respective compartments on signalling as well as axon retraction by Ephrin-A5 and Slit1 and also axon guidance in vivo. The combination of these molecular tools makes this an elegant study that demonstrates how different localised changes in intracellular signalling molecules can contribute to different cellular effects.

We thank the reviewer for her/his very encouraging evaluation of our work.

Comments:

1. The quantitative interpretation of the traces must represent a challenge, given the relatively low signal compared to the noise. However, there is no description in the manuscript about how the traces are interpreted, except that the data depict the amplitude (according to the figure legends).

Please provide details about how the amplitude is measured. If relying on the amplitude after stimulation gives large variability, perhaps the authors could consider if measuring AUC (area under the curve) after agonist application would be a more robust measure of changes in FRET?

We agree that the amplitude measurement is quite variable. This might be linked to the level of expression of the guidance cue receptor. For instance, each retinal ganglion cell expresses an amount of EphAs related to the position of its cell body in the retina: there is a temporo-nasal gradient of EphA expression in the retina. In our *in vitro* preparation, we cannot determine the position of the axon cell body in the retina and thus cannot extrapolate how many EphA receptors it carries. This might be a source of the observed variability.

We now provide further explanation about how the amplitude of the FRET signal is measured in the method section (line 965). Following the suggestion of the reviewer, we measured AUC to try to provide a more robust measure of the FRET changes. We found that this quantification method does improve the dataset and thus chose not to show it to avoid increasing further the density and complexity of the figures.

2. What is the basis for the mean±SEM in the traces as opposed to the data included in the box/whisker plots? Please write this clearly. E.g. are the traces based on several cells from the same slide, whereas the box/whisker plots are from all the cells from all the slides? Frequently the mean ± SEM data in the traces do not seem to match well the data in the box/whisker plots, e.g., in figure 2a (cytosolic H147 sensor), where the Slit1-induced mean change in FRET as read from the traces indicates a ~5% decrease, whereas the box/whisker plot indicates a ~2% increase, so please clarify.

We apologize for the lack of clarity in the description of the analysis methods, we added an extended description of the analysis pipelines in the methods. Briefly, the CFP and YFP intensity were first measured from the raw images and corrected for background and bleedthrough from the CFP into the YFP channel. The FRET ratio was computed from these corrected intensity measurements. The traces obtained exhibited a drift leading to an increase of the measured ratio that was obvious before the addition of any axon guidance molecule and is likely due to photobleaching. Because of this observation we chose to perform two parallel analysis pipelines.

To avoid distorting the dataset distribution for statistical analysis (*i.e.* box&whisker plots), we measured $\Delta R/R_0$ from the raw data, using the following formula: $\frac{\Delta R}{R_0} = \frac{\bar{R}_{[11-12min]} - \bar{R}_{[9-10min]}}{\bar{R}_{[9-10min]}}$. Still the drift in the ratio was present in this dataset and is represented by the increase of the amplitude detected in the sham experiment (PBS). Thus, whether the signal is an increase or a decrease should be read as a comparison with the sham experiment measurement.

To facilitate the interpretation for the reader, we corrected the traces for the progressive drift observed before applying ephrinA5, Slit1 or PBS. This correction is based on a linear fit of the measurement before the stimulation. The traces presented in the figures include this correction, thus the reader can see an elevation (or a decrease) of the cyclic nucleotide concentration from a single trace, without having to compare with the sham stimulation.

These two parallel analysis pipelines have been clarified in the method section (line 965). They explain most of the difference between the traces and the box&whiskers graph. An additional contribution might come from the distinct central tendency used. We use the median and interquartile ranges for box & whisker plots (once again because it is more adapted than the mean for statistical analysis and distribution representation since it is less influenced by potential outliers) and

mean +/- sem for traces (easier to interpret for a graphical analysis). The central tendency used is mentioned in each figure legend.

Overall, we favor a dual representation to help the reader for a quick grasp of the meaning of the figures. Both representations are anyway consistent for all figures (when comparing with the sham experiment for box&whiskers plot).

3. On the same note, in figure 2a (H147-Kras): The average traces shows first a decrease then an increase? Which value is reported as the amplitude? The maximal negative value? This is important, as the average in the box/whisker is 0, although the trace seems to indicate that there is a reduction in FRET (ie. a reduction in cAMP).

The comment of the reviewer reflects the dual analysis pipeline explained in the response to his previous points. We think that this is now clarified with the additional explanation provided in the method section.

4. For some biosensors (for example epac (ancestor of H147 biosensor used in the current manuscript)), adding a targeting sequence decreases the maximal efficacy of the biosensor (DOI: 10.1038/ncomms15031), limiting the comparison across different biosensors. Does the Lyn- or Kras-H147 show altered efficacy compared to cytosolic H147 (from the average traces, the maximal response (Fsk) seems reduced for the Lyn-H147)? If so, would it be more appropriate to present FRET signal as a percent of maximal FRET obtained with Fsk in figure 2a?

We agree with the reviewer that adding a targeting sequence has been previously shown to potentially modify the behavior of biosensors. This is the reason why we never compared biosensors that do not carry the same targeting sequence. All comparisons were done between experiments targeting the same compartment. Still, one might argue that the sensitivity of one of the targeted biosensors might not be sufficient to detect a signal and that the lack of change in FRET might not reflect an absence of second messenger modulation but rather a poor sensitivity of the targeted biosensor. This is not the case in our study since all targeted biosensors have been able to detect a guidance cue-induced second messenger modulation. This is reported either in the present manuscript or, for Lyn-H147, in Averaimo et al. (Nat. Commun, 2016). The latter study has been performed using the same microscopy setup as the present manuscript.

We understand the rationale of normalizing the Slit1-induced change by the Fsk-induced FRET variation. However, targeting sequences might alter both the maximal FRET change and the affinity of the sensor for its ligand. Since we feel that an extensive biochemical characterization of the sensors is outside of the scope of this study, we chose to show the data as raw as possible, thus avoiding to normalize to the maximal FRET response.

5. The targeted twitch biosensors developed in the current manuscript would be valuable tools for the biosensor community. Related to the questions on the targeted H147 biosensors (above), does the targeting affect the maximal efficacy of the Ca²⁺ biosensors? How would this affect the measurements?

We again agree with the referee that Twitch2b maximal efficacy might be affected by the targeting. Like for H147, we did conduct an extensive biochemical characterization of the targeted Ca²⁺ sensors. For the same reasons as above (all targeted Ca²⁺ sensors are able to detect Ca²⁺ transients in at

least one experimental condition), we think that the measurements are not affected by a potential difference in the FRET maximal response between the lipid raft-targeted and -excluded biosensors. In addition, for Ca²⁺ transient analysis, the quantification were done using the frequency of the Ca²⁺ transient as a proxy. The frequency is much less prone to variation in the amplitude since it does not rely directly on an amplitude measurement.

6. There are still some mechanistic aspects that are not clear and that should at least be discussed, e.g.: How does Ephrin-A5 and Slit1 reduce cAMP? How is reduced cAMP linked to reduced Ca²⁺?

Although the mechanistic of the regulations mentioned by the reviewer has not been identified, we now include in the discussion a potential molecular explanation of the interaction between second messengers (line 369).

Minor:

Some of the figures would be easier to read if the sensor used were indicated (as already done in some of the figures)

There is inconsistency in the spelling of SponGee vs. Spongee in the figures

Line 581: missing R in Kras

Line 787-789: subscript lacking in formulas

Line 794-795: please check final concentrations of spermine-NONOate and forskolin that cells were exposed to. 50mM and 10mM seem more like stock concentrations and could lead to off-target effects if cells are exposed to these concentrations.

HEK293 cells: described in M&M, not clear which experiments were performed in HEK293 cells.

We thank the reviewer for her/his careful reading. The typo and mistakes mentioned have been fixed according to the reviewer suggestions.

Reviewer #3 (Remarks to the Author):

In this manuscript, Baudet et al. have investigated second messenger signaling in subcellular compartments, so-called raft and non-raft membrane microdomains, in retinal ganglion cell axons responding to the repulsive cues ephrin-A5 and Slit1. Using latest tools to monitor or scavenge cAMP, cGMP and Ca²⁺ in these subcellular compartments, the authors showed that ephrin-A5 and Slit1 exert their repulsive activities through distinct signaling networks in raft and non-raft microdomains, respectively. I think that their findings substantially elaborate our understanding of molecular signaling underlying axon guidance. Below are my comments that I believe will help improve the quality of this work.

We thank the reviewer for her/his positive assessment of our work.

1. The authors have stated, e.g., in the abstract, that ephrin-A5 and Slit1 activate subcellular-specific second messenger crosstalks, each signaling network controlling distinct axonal morphology changes in vitro and pathfinding decisions in vivo. However, it is unclear whether the observed difference in axonal responses in vitro (Fig. 5) depends on subcellular localization of second messenger crosstalks.

We agree with the reviewer that the relationship between the observed difference in axonal responses *in vitro* and the subcellular localization of second messenger was not fully demonstrated. We conducted optogenetic experiments that now demonstrate this link. These experiments are illustrated in Fig. 8.

Also, the in vivo data in Fig. 6 are difficult to interpret because the reagents used in these experiments can perturb second messengers that are unrelated to the downstream signals of ephrin-A5 and Slit1.

We agree with the reviewer that altering second messenger signaling might perturb more than only ephrin-A5- or Slit1-dependent axon pathfinding. We anyway provide some extent of specificity by restricting the buffering of cAMP, cGMP and Ca²⁺ to subcellular compartment. It might not be enough to achieve perfect specificity. We toned down our conclusion (line 330). To support the link between ephrin-A5- or Slit1-dependent axon pathfinding and local networks of second messengers, we also complemented the manuscript with the set of optogenetic manipulation of cyclic nucleotides described in Fig. 8, demonstrating that local changes of these signaling molecules in and outside lipid rafts mimic the axonal behavior induced by ephrin-A5 and Slit1, respectively.

2. Although it is concluded in line 130 that Twitch2b-Kras is not targeted to lipid rafts (Supplementary Fig. 1), the immunoblots in this figure actually indicate the presence of Twitch2b-Kras in raft fractions, e.g., a substantial amount of Twitch2b-Kras in the fraction 4 colocalizing with caveolin.

We agree with the reviewer that the membrane fractionation exhibits lipid raft-excluded proteins in the supposed raft fraction, in particular in fraction 4. However, this observation is valid for both Twitch2b-Kras and Adaptin, the latter being a protein already validated as a raft-excluded protein. Since the fraction pattern of Twitch2b-Kras and Adaptin are very similar (illustration in Figure S1), we think we can claim that Twitch2b is excluded from lipid rafts. This is confirmed by the control of the spatial specificity of the different sponges shown in Fig S2

3. The data in Figure 3 are consistent with the proposed models of signaling networks (d and h) but do not substantiate the validity of these models. The authors should tone down their description about these signaling networks.

To substantiate the validity of the proposed models of signaling networks, we conducted optogenetic experiments using lipid raft-targeted and -excluded light activatable adenylyl and guanylyl cyclases (Fig. 8). The obtained results are again consistent with the proposed models of signaling network, but might still leave room for other interpretations. We thus toned down our conclusions and acknowledge that the requirement of all interactions between second messengers for axon retraction is not directly demonstrated (line 296).

4. How do ephrin-A5 and Slit1 elicit subcellular-specific second messenger signaling? Are receptors of ephrin-A5 and Slit1 localized to raft and non-raft domains of growth cone plasma membranes, respectively?

How ephrin-A5 and Slit1 elicit subcellular-specific second messenger signaling is unclear. EphAs, the ephrin-A receptors have been shown to be able to shuttle between lipid rafts and the non-raft-fraction of the plasma membrane. Similarly, the domain of the membrane containing Robo1 and Robo2 has not been identified. We attempted to localize the subcellular domain containing EphA and

Robo receptors in the retina. Although the membrane fractionation method was efficient for other proteins (see Fig. S1), we did not succeed in obtaining readable western-blot after membrane fractionation of retinal extracts using a range of antibodies (EphA4 (4C8H5) 37-1600 (Invitrogen), EphA5 PA5 89694 (ThermoFisher), EphA6 AER(016) (Alomone labs), Robo1 AF1749 RDSsystems, Robo1 20219-1-AP (Proteintech) and a Robo2 antibody developed by the lab of Alain Chédotal). Since this attempt did not succeed we now mention in the discussion that “The spatial specificity of each pathway might be related to the subcellular localization of the receptors, although the subcellular location of EphA and Robos is unclear” (line 366).

Reviewer #4 (Remarks to the Author):

The present manuscript addresses the question of how second messengers discriminate between different repellent axon guidance cues. The authors use genetically-encoded scavengers to define the subcellular compartmentation of cyclic nucleotides (cAMP and cGMP) and Ca²⁺ signals to define the response of retinal growth cones to Slit1 and ephrin-A5, two repellent axon guidance molecules that are known to influence these axons along their pathway up to the superior colliculus. They show that second messenger signaling for ephrin-A5 and Slit1 and their interaction occurs in two distinct submembrane compartments: lipid rafts and the non-raft domain of the plasma membrane. They further used in vitro and in vivo approaches to show that altering the subcellular distribution of the second messengers induces axon guidance defects. They conclude that cAMP, cGMP and Ca²⁺ do not really act as molecular integrators of axon guidance cues, as previously thought, but rather provide distinct information in part encoded by different subcellular localization.

This is an interesting study that offers conceptual advances related to the mechanism that allow a growth to interpret different repellent guidance cues. The study is in general well conducted.

We thank the reviewer for Her/his highly positive assessment of our study

I have a few comments that may help improving the study:

1. Figure 1-3 are rather packed and difficult to follow. Can the authors display the items in a clear way? Can they also show a few examples of what are the actual images they use to obtain the different graphs? This will help the reader to understand better how the data were obtained.

We thank the reviewer for her/his suggestion and apologize for the packed figures. We split Fig. 2, 3 (former numbers) to ease the reading of these figures and now provide examples of images for each second messenger imaged (Fig. 1 and 2). In addition the biosensor used has been indicated in each figure panel.

2. All the graphs related to the frequency of Ca²⁺ transients show a variability within the same condition and several samples do not seem to have any response. What is the explanation for this variability?

We agree that the measurements are quite variable. This might be linked to the level of expression of the guidance cue receptor. For instance, each retinal ganglion cell expresses an amount of EphAs

related to the position of its cell body in the retina: there is a temporo-nasal gradient of EphA expression in the retina. In our *in vitro* preparation, we cannot determine the position of the axon cell body in the retina and thus cannot extrapolate how many EphA receptors it carries. This might be a source of the observed variability. Previous studies monitoring Ca²⁺ transients also report variability that might be linked to the neuronal subtype (Gomez et al., Science 1999). Since retinal ganglion cells is a quite heterogeneous group of neurons, it is possible that different subtype of retinal ganglion cells exhibit different Ca²⁺ transient frequency. It is however unlikely to affect our conclusion since we monitor the same cell before and after the exposure to ephrin-A5 or Slit1.

3. The in vivo data show limited defects in few of the electroporated axons, why?

We agree with the reviewer that we detected only a moderate number of axons exhibiting a defect in axon pathfinding. The pathfinding of these axons not only rely on a single axon guidance family but is regulated by other axon guidance molecules by cell adhesion molecules and by the extracellular matrix. We think that it explains the moderate number of guidance defects. This is also observed for instance in Slit knock-out animals that also exhibit limited pathfinding defect at the chiasm (Plump et al. 2002, Thompson et al. 2006, ref 31 and 32 of the manuscript). This explanation is now included in the manuscript (line 334).

4. Slit1 controls intra-retinal axons guidance. Have the authors check if there is a retinal phenotype in the different conditions

The pigmentation of the retinal pigmented epithelium prevented us to track intra-retinal axon pathfinding using light-sheet microscopy since the samples were bleached to eliminate the pigments. However, to complement our initial observations, we analyzed another phenotype observed in Slit-deficient animal. These animals exhibit an enhanced number of axons growing in the contralateral optic nerve (Plump et al. 2002, ref 31 of the manuscript). Similarly, the animals electroporated with SpiCee-Kras and SponGee-Kras also show extra retino-retinal axons. This is now illustrated in Fig. 9.

5. The text is not always easy to follow and some sentences may benefit from some simplification

We apologize for the complicated sentences. We checked the text of our manuscript to simplify the lengthy and complex sentences.

REVIEWERS' COMMENTS

Reviewer #2 (Remarks to the Author):

"We now provide further explanation about how the amplitude of the FRET signal is measured in the method section (line 965). Following the suggestion of the reviewer, we measured AUC to try to provide a more robust measure of the FRET changes. We found that this quantification method does improve the dataset and thus chose not to show it to avoid increasing further the density and complexity of the figures."

Thank you for describing how the peaks were analyzed. If measuring AUC improved the dataset, then I am not sure why it was not implemented. Do the authors mean the opposite, i.e. "We found that this quantification method does "NOT" improve the dataset"?

Point 2:

Thank you for clarifying that the data used to generate the box/whisker plots were corrected for the baseline drift. This helps explaining apparent discrepancies between tracings and box-whisker plots, such as in Figure 2b H147-Kras.

Statistics: The authors use non-parametric statistics, which is well suited for data that are not normally-distributed. However, when several groups are compared (such as Fig1b, 3a, 4a, 4c, 5a, 5c, 6a-e, and several supplementary figures), the authors use the Kruskal-Wallis test followed by Mann-Whitney post hoc test, but from the figure legend or methods it is not clear whether the authors conducted a correction for making multiple comparisons (e.g. Bonferroni or Dunn's multiple comparisons test), which would be needed to avoid type I errors.

Point 5:

I agree with the authors that frequency of Ca²⁺ transients are less prone to changes in maximal efficacy of the Ca²⁺ biosensors.

These targeted biosensors will be of great value to the biosensor community. If all targeted Twitch biosensors have been extensively characterized, please provide such data or a reference to the characterization of these biosensors.

Minor:

Concerning valve opening (line 970-1): please indicate how cells were superfused.

Reviewer #3 (Remarks to the Author):

This manuscript has been substantially revised according to my previous comments, and I am satisfied with the current version.

Reviewer #4 (Remarks to the Author):

The authors have addressed most criticisms. The manuscript provides important information and it will be an important addition to the field of axon guidance

We thank the reviewers for the kind assessment of our work.

Reviewer #2 (Remarks to the Author):

“We now provide further explanation about how the amplitude of the FRET signal is measured in the method section (line 965). Following the suggestion of the reviewer, we measured AUC to try to provide a more robust measure of the FRET changes. We found that this quantification method does improve the dataset and thus chose not to show it to avoid increasing further the density and complexity of the figures.”

Thank you for describing how the peaks were analyzed. If measuring AUC improved the dataset, then I am not sure why it was not implemented. Do the authors mean the opposite, i.e. “We found that this quantification method does “NOT” improve the dataset”?

Yes, indeed. We meant “We found that this quantification method does NOT improve the dataset”. This is the reason why AUC was not implemented. We apologize for this typo in the response to the reviewer file.

Point 2:

Thank you for clarifying that the data used to generate the box/whisker plots were corrected for the baseline drift. This helps explaining apparent discrepancies between tracings and box-whisker plots, such as in Figure 2b H147-Kras.

Statistics: The authors use non-parametric statistics, which is well suited for data that are not normally-distributed. However, when several groups are compared (such as Fig1b, 3a, 4a, 4c, 5a, 5c, 6a-e, and several supplementary figures), the authors use the Kruskal-Wallis test followed by Mann-Whitney post hoc test, but from the figure legend or methods it is not clear whether the authors conducted a correction for making multiple comparisons (e.g. Bonferroni or Dunn’s multiple comparisons test), which would be needed to avoid type I errors.

The post-hoc test implemented when using a Kruskal-Wallis test indeed includes a multiple comparison correction (Dunn’s multiple comparison test). This is now clarified in the figure legends.

Point 5:

I agree with the authors that frequency of Ca²⁺ transients are less prone to changes in maximal efficacy of the Ca²⁺ biosensors.

These targeted biosensors will be of great value to the biosensor community. If all targeted Twitch biosensors have been extensively characterized, please provide such data or a reference to the characterization of these biosensors.

The untargeted biosensor has been described extensively in Thestrup et al 2014. The subcellular targeting of Lyn-Twitch2b and Twitch2b-Kras are described in our study (Figure S1). However, further biochemical characterization of the targeted Twitch2b (Kd, EC50, ...) are not available. As agreed by the reviewer, it does not affect the ability to use these sensors for our study, although it would be useful for others. The above-mentioned reference is included in the manuscript (ref 36).

Minor:

Concerning valve opening (line 970-1): please indicate how cells were superfused.

Retinal explants were superfused using a close chamber and a syringe-pump, to avoid imaging artefacts generated by the pulses of peristaltic pumps. These further details have been included in the Methods.

Reviewer #3 (Remarks to the Author):

This manuscript has been substantially revised according to my previous comments, and I am satisfied with the current version.

We thank the reviewer for her/his positive evaluation of the manuscript improvement during the revision process.

Reviewer #4 (Remarks to the Author):

The authors have addressed most criticisms. The manuscript provides important information and it will be an important addition to the field of axon guidance

We thank the reviewer for her/his kind assessment of our work and its impact.